# GOAT: A Training Framework for Goal-Oriented Agent with Tools

## Abstract

Large language models (LLMs) have recently been extended beyond traditional text generation to serve as interactive agents capable of using external tools based on user intent. However, current LLM agents still show limited ability to handle *goal-oriented* queries, which require decomposing a high-level objective into multiple interdependent API calls with correct planning and execution. Current approaches mainly rely on zero-shot evaluation due to the absence of training data. While proprietary closed-source models such as GPT-4 demonstrate strong reasoning abilities, smaller open-source models struggle to perform complex tool use effectively. Thus, we propose a novel training framework **GOAT**, which enables fine-tuning of LLM agents in a human annotation-free setting. GOAT automatically constructs synthetic datasets of goal-oriented API execution tasks directly from given API documents, equipping models with the ability to reason over interdependent calls and generate coherent responses. Through extensive experiments, we show that GOAT-trained agents achieve state-of-the-art performance across multiple existing goal-oriented benchmarks. In addition, we introduce **GOATBench**, a new goal-oriented API execution benchmark, and demonstrate that agents trained with GOAT also excel in this setting. These results highlight GOAT as a practical path toward building robust open-source LLM agents capable of complex reasoning and tool use.

## 1 Introduction

Recent advances in large language models (LLMs) have led to remarkable progress across a wide range of natural language processing tasks (Zhao et al., 2025; Achiam et al., 2023). Beyond their success in pure language understanding, a growing line of research explores a new paradigm, where LLMs function as *agents* that can actively interact with the external world by connecting to tools such as APIs to respond to user needs (Qin et al., 2024; Patil et al., 2023; Schick et al., 2023). This paradigm, referred to as *tool learning*, highlights the shift from treating LLMs purely as text generators to enabling them to plan tasks, invoke external tools, and provide answers to user queries.

While prior literature on tool learning includes relatively simple tasks such as single-step queries (Patil et al., 2023; Schick et al., 2023) or fine-grained queries with explicitly specified instructions (Qin et al., 2023; Liu et al., 2024; Shen et al., 2023b), we focus on more realistic and challenging scenarios, which we term as **goal-oriented tasks**. As illustrated in Figure 1, a goal-oriented user query provides only a high-level objective of the task rather than detailed step-by-step instructions, requiring the agent to break down the objective into intermediate steps, determine which APIs to call, and infer appropriate arguments for the selected API functions. Consequently, these tasks demand strong reasoning for task decomposition, long-horizon planning, and call realization that captures the interdependency between APIs, followed by their coordinated execution.

However, progress on such goal-oriented tasks has been limited by the lack of training data: constructing datasets that capture inter-API dependencies needs human annotation and is prohibitively costly at scale. Consequently, current approach (Song et al., 2023) mainly rely on zero-shot evaluation, where models are expected to perform complex reasoning without task-specific supervision. To address these challenges, we propose a novel training framework for **G**oal-**O**riented **A**gent with **T**ools **(GOAT)**, which enables fine-tuning LLM agents even in the absence of human-annotated data through a fully automatic synthetic data generation pipeline. Our approach builds on the common

Figure 1: **Goal-oriented API execution task.** To solve a goal-oriented user query, the LLM agent performs step-by-step task planning, executes a sequence of interdependent API calls, and generates a natural language response. The figure illustrates the workflow where the user query is decomposed into subtasks, mapped to API calls, and each function call is executed by filling API arguments based on the outputs of previous responses, ultimately yielding a coherent answer.

practice that once a target set of APIs is specified, their function documentation is already available, which can be directly leveraged to construct training data. Specifically, given API documentations of a target environment, GOAT first induces an API dependency graph through a refinement pipeline, retaining only feasible invocation relations between APIs. From this graph, GOAT samples connected subgraphs, representing interdependent subtasks of a goal-oriented task. These subgraphs are converted into training samples by instantiating and executing the corresponding APIs. GOAT then generates a goal-oriented user query and a final response aligned with the resulting API sequence. Finally, GOAT jointly fine-tunes the LLM and the retrieval model on the resulting dataset, equipping them with the ability to reason over interdependent APIs and produce coherent responses.

Furthermore, leveraging our data generation pipeline together with human labeling, we curate **GOATBench**, an evaluation benchmark for goal-oriented tasks. Through extensive experiments on goal-oriented benchmarks including GOATBench, we show that GOAT enables open-source models to achieve state-of-the-art performance in multiple benchmarks, in some cases even surpassing certain closed-source models with strong reasoning capabilities. Our main contributions are as follows:

- We propose **GOAT**, a novel training framework that automatically constructs goal-oriented API execution datasets from the target API documents without human annotation, enabling efficient domain-specific adaptation of LLM agents and strengthening their reasoning capabilities.
- Through extensive experiments on goal-oriented benchmarks, we demonstrate that agents trained with GOAT achieve state-of-the-art performance with open-source models.
- We further introduce **GOATBench**, a new evaluation benchmark for goal-oriented tasks, and confirm consistent performance gains of GOAT-trained agents on this benchmark as well.

## 2 RELATED WORK

**Task Formulations in Tool Learning**  Tool learning tasks can be categorized by the level of reasoning required between user queries and API calls. In the simplest setting, a few APIs are directly provided in the prompt, so models learn only to operate within this restricted set in straightforward ways, making the task considerably easier and more constrained (Yao et al., 2023; Schick et al., 2023; Surís et al., 2023; Zhuang et al., 2023). A more advanced formulation arises when many tools are available, requiring retrieval of the relevant one. Among these, some tasks provide queries that explicitly specify each step-by-step instructions, leaving little need for planning and reasoning (Patil et al., 2023; Qin et al., 2023; Shen et al., 2023a; Yang et al., 2023; Shen et al., 2023b; Liu et al., 2024). In contrast, the most challenging and realistic setting, which we target in this work, is the *goal-oriented task*, where user queries describe only high-level goals rather than explicit execution steps. Here, the system must plan and execute a sequence of interdependent API calls to address the query holistically (Song et al., 2023; Li et al., 2023). While closed-source models have achieved promising results in this setting, absence of training data makes boosting the performance of open-source models challenging. Our work addresses this gap by introducing a training framework designed to handle realistic, goal-oriented tasks.

Table 1: **Comparison of existing synthetic training data generation works in tool learning.** Our work uniquely generates synthetic training data specifically targeting goal-oriented tasks. *Note that although API-Bank provides a benchmark that includes goal-oriented tasks, its training data generation process does not target such queries.

| Work | Real API | Fully Automatic | Scalable | API Call Dependency | Goal-oriented Query |
|------|----------|-----------------|----------|---------------------|---------------------|
| ToolFormer (Schick et al., 2023) | ✗ | ✓ | ✗ | ✗ | ✗ |
| Gorilla (Patil et al., 2023) | ✓ | ✓ | ✓ | ✗ | ✗ |
| ToolLLM (Qin et al., 2023) | ✓ | ✓ | ✓ | ✗ | ✗ |
| API-Bank* (Li et al., 2023) | ✗ | ✓ | ✗ | ✗ | ✗ |
| TaskBench (Shen et al., 2023b) | ✓ | ✗ | ✗ | ✓ | ✗ |
| ToolFlow (Wang et al., 2024b) | ✓ | ✓ | ✓ | ✓ | ✗ |
| Magnet (Yin et al., 2025) | ✓ | ✓ | ✓ | ✓ | ✗ |
| ToolDial (Shim et al., 2025) | ✓ | ✓ | ✓ | ✓ | ✗ |
| **Ours (GOAT)** | ✓ | ✓ | ✓ | ✓ | ✓ |

**Synthetic Training Data for Tool Learning** High-quality training data is essential for developing LLM agents that can reliably use tools in real-world scenarios. Earlier efforts generated single-API training data by first selecting an API function and then constructing a corresponding query–api call pair (Schick et al., 2023; Patil et al., 2023). Subsequent work expanded to multi-API settings by randomly sampling multiple APIs, generating a combined query and the corresponding API call path, which resulted in parallel step-by-step instructions rather than high-level goals (Qin et al., 2023; Li et al., 2023). Graph based approaches build API dependency graph to sample connected API sequences, but mainly to yield either fine-grained instructions (Shen et al., 2023b) or multi-turn dialogues (Wang et al., 2024b; Yin et al., 2025; Shim et al., 2025), with each node corresponding to a single API call. The dependencies modeled in these graph-based frameworks are relatively weak, as they are derived from simple heuristics or algorithms without validating whether the connections are truly meaningful, and none of the above generate holistic goal-oriented queries that require agents to plan over multiple interdependent API calls. In contrast, our approach automatically constructs such goal-oriented data, thereby addressing this gap and enabling effective training for realistic tool-use scenarios (see Table 1).

## 3 A TRAINING FRAMEWORK FOR GOAL-ORIENTED AGENT WITH TOOLS

We propose a novel human-annotation-free training framework for Goal-Oriented Agents with Tools (GOAT), motivated by the prohibitive cost of collecting manual annotations. In contrast to prior works that primarily rely on zero-shot inference with pretrained LLMs alone, we begin with the practical observation that agents are typically deployed in specific domains with a fixed set of target APIs. This makes it reasonable to assume that API documentation for these APIs is available beforehand. Leveraging this assumption, our framework (i) automatically generates training samples from the available API documentation, and (ii) fine-tunes an LLM and retriever model on these samples to strengthen its goal-oriented reasoning capabilities for the target API environment.

### 3.1 AUTOMATIC DATASET CONSTRUCTION

A central challenge in training goal-oriented agents is the absence of manually annotated training datasets, which prevents models from acquiring task-specific supervision. Our framework addresses this challenge by fully automatically constructing synthetic training data, thereby eliminating the need for costly human annotations. It is important to note that intermediate API calls in goal-oriented tasks are inherently interdependent, as earlier calls are often executed to prepare inputs for later ones. In our framework, the generated synthetic data must therefore capture this interdependency across API calls. Such dependency information is typically implicit in API documentation, and we leverage this dependency cues during the data generation process. Specifically, our method constructs goal-oriented API execution data through two main stages. First, given a set of API documents describing API functions, we build an API dependency graph that captures all possible ways in which the output of one API can serve as an input to another (Section 3.1.1). From the resulting graph, we extract connected subgraphs to create synthetic data points, each consisting of a goal-oriented user query $u$, a set of call units $\{(s, c, o)\}$—each consisting of a sub-query $s$, the corresponding API call $c$, and its output $o$—and the final response $r$ (Section 3.1.2).

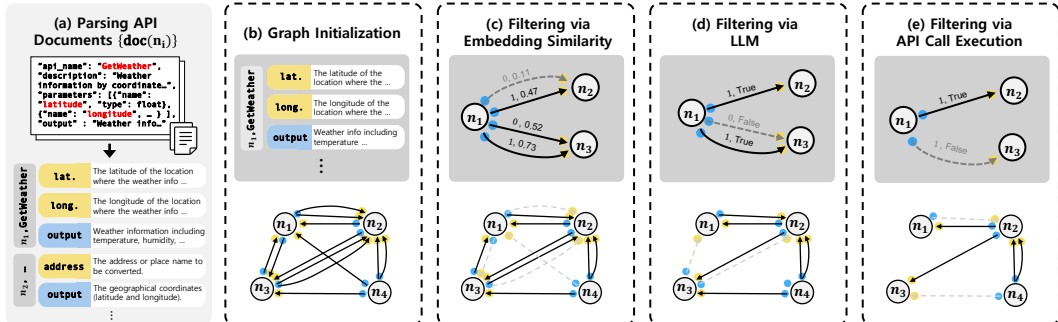

Figure 2: **The overview of API dependency graph construction process.** Given the API documents, each document is first parsed to extract function descriptions, which are then used to initialize a raw dependency graph in (a). This graph is progressively refined through three filtering steps (c)-(e), resulting in the final API dependency graph that captures reliable relations among APIs. The graphs shown under (b)-(e) illustrate how the API dependency graph evolves as it is progressively refined through each filtering step.

### 3.1.1 API DEPENDENCY GRAPH CONSTRUCTION

Given a set of API documents $\{\text{doc}(n_i)\}$ describing API functions $n_i$ with implicit dependency information, we formulate an API dependency graph $G = (\mathcal{V}, \mathcal{E})$, where $\mathcal{V} = \{n_i\}$ represents the set of API functions and an edge in $\mathcal{E} = \{(n_i, n_j, k)\}$ indicates that the output of an API function $n_i$ can be used as the $k$-th argument in a subsequent call to $n_j$. Since a single output may serve multiple parameters for the same API function, $G$ is a multidigraph that allows multiple directed edges between the same pair of nodes. The dependency graph captures the input–output dynamics across APIs, reflecting the execution flow and forming the basis for goal-oriented workflows.

To construct a correct dependency graph $G$, we leverage LLMs for their strong reasoning capabilities: a high-performing LLM generates candidate arguments for a source API, executes them, and verifies whether the resulting output can populate the parameters of a destination API. However, performing this procedure exhaustively across all input–output pairs is prohibitively expensive. To balance reliability and efficiency, we instead begin with an over-complete graph and progressively reduce LLM usage through a staged filtering pipeline: inexpensive coarse filters eliminate clearly incompatible edges, while only a shrinking subset is escalated to increasingly precise (and costlier) checks. The effect of each stage is detailed in Appendix A.2. The overall process is illustrated in Figure 2, and each stage is described in detail in the following.

**API Document Parsing and Graph Initialization**    We begin by extracting the input and output specifications of each API function $n_i$ by parsing its corresponding documentation $\text{doc}(n_i)$, which define the endpoints for edges in the graph (see Figure 2a). In particular, each specification includes the natural language description of parameters and output; we denote the description of the $k$-th input of function $n_i$ as $\text{In}(n_i, k)$ and the output description of $n_i$ as $\text{Out}(n_i)$. The extraction is accomplished by prompting an LLM with the template described in Appendix I.1, and the resulting structured representations are illustrated at the bottom of Figure 2a and detailed further in Appendix A.1. Using these extracted specifications, we then construct an initial fully connected multidigraph by adding directed edges from the output of each API function to every input parameter of every other API function (see Figure 2b). This over-complete graph tentatively connects all possible API pairs and serves as the starting point for subsequent filtering and refinement.

**Filtering via Embedding Similarity**    To prune unlikely edges, we compare the descriptions of source API outputs and destination API input obtained from the previous step. For each edge $(n_i, n_j, k)$, we compute the cosine similarity between Sentence-BERT (SBERT) (Reimers & Gurevych, 2019) embeddings of $\text{In}(n_j, k)$ and $\text{Out}(n_i)$. Edges with similarity scores below a threshold $\tau$ are discarded. Note that since this method does not require invoking an LLM, it provides a very efficient way to filter out clearly incompatible pairs. At the same time, because more precise filtering will be applied in later stages, we set $\tau$ with a low threshold—favoring recall over precision—to avoid over-pruning and to retain most potentially valid connections. This filtering step is illustrated in Figure 2c.

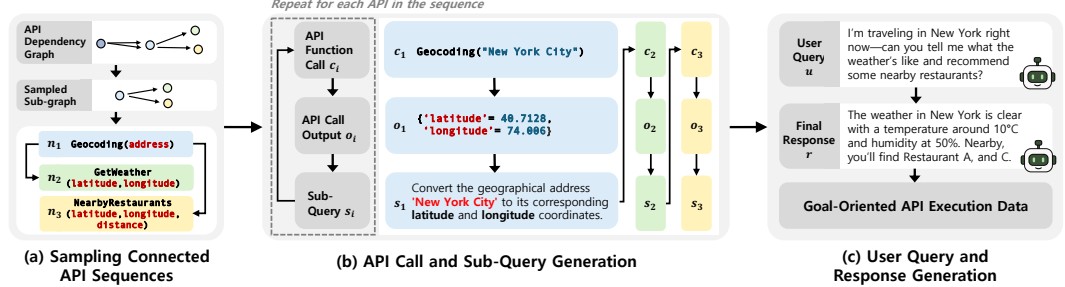

(a) Sampling Connected API Sequences    (b) API Call and Sub-Query Generation    (c) User Query and Response Generation

Figure 3: **Overview of goal-oriented API execution data construction.** The process involves (a) sampling connected API sequences, (b) generating API calls, outputs, and sub-queries, and (c) composing user queries and final responses.

**Filtering via LLM**  As illustrated in Figure 2d, edges that survive the similarity filter are further examined leveraging the reasoning capability of LLMs with a single LLM call per edge. Given API documents ($\text{doc}(n_i)$ and $\text{doc}(n_j)$) and the descriptions ($\text{In}(n_j, k)$ and $\text{Out}(n_i)$), the LLM determines whether the output of $n_i$ can meaningfully populate the $k$-th input of $n_j$ using the prompt in Appendix I.2. When the LLM judges the validity of edges, the LLM also generates a natural-language justification explaining why an edge is valid for edges deemed valid. We reuse these justifications later in data construction (Section 3.1.2) to guide argument generation. This filtering step checks for *semantic plausibility* based on descriptions, without grounding in actual values.

**Filtering via API Call Execution**  As illustrated in Figure 2e, candidate edges are finally validated through execution with three LLM calls per edge. Unlike the previous semantic-only check, this step grounds edges in *concrete values* from real API executions. For each edge $(n_i, n_j, k)$, the LLM instantiates a call $c_i$ for $n_i$ by generating plausible arguments and execute it to obtain output $o_i$. It then constructs a destination API's call $c_j$ for $n_j$, filling the $k$-th argument with content extracted from $o_i$, and finally verifies whether $c_j$ is coherent and executable as a continuation of $c_i$ (Appendix I.3). By validating edges through real outputs, we eliminate connections that are semantically plausible but non-executable, ensuring that the final dependency graph represents reliable API workflows.

### 3.1.2 GOAL-ORIENTED API EXECUTION DATA CONSTRUCTION

Once the dependency graph is finalized, we construct goal-oriented task samples where an user query leads through a sequence of interdependent API calls, to a final response. This process consists of three main steps. First, since the dependency graph is composed of all possible API relationships, we extract connected subgraphs and treat them as candidate API sequences. Second, for each sequence, we sequentially instantiate and execute each API functions by filling in their input arguments, executing them in order, and passing API outputs forward as inputs to subsequent calls. In doing so, we also generate sub-queries that explain each API call in natural language. Finally, based on the sub-queries and the outputs obtained from calling a API sequence, we construct a user query that captures the overall intent, and then generate a natural final response that interprets the outputs in their full context to address this query. The result is a dataset of samples where every component— from query to intermediate reasoning to final response—is grounded in real API executions. The full process is illustrated in Figure 3.

**Sampling Connected API Sequences**  As illustrated in Figure 3a, we begin by extracting all possible *connected* subgraphs from $G$, with up to $L = 4$ nodes, each of which serves as the basis of a task instance. If the subgraph contains cycles, we randomly break edges, producing an acyclic structure. The resulting acyclic multidigraph is then topologically sorted to produce an execution order $(n_{k_1}, n_{k_2}, \ldots, n_{k_L})$ where $k_l$ represents the $l$-th node index among the $L$ nodes. This guarantees that APIs are invoked in a dependency-respecting sequence, forming a valid workflow.

**API Call and Sub-Query Generation**  Following the predetermined sequence, each $n_{k_\ell}$ is instantiated into an API call $c_{k_\ell}$ by filling the required input arguments (Appendix I.4). For each parameter, there are two cases: (i) when the subgraph contains an edge leading to it, where the argument should be obtained from the output $o_{k_m}$ of a prior API ($m < \ell$), and (ii) when no such edge exists. In the latter case, the LLM refers to the function's specification in $\text{doc}(n_{k_\ell})$ to synthesize a plausible value.

In the former case, the result $o_{k_m}$ is provided, and the LLM extracts appropriate fields to populate the argument. Since outputs are often complex and nested, non-trivial reasoning over the execution context is required; here, we prompt the LLM with the justification produced during the second filtering stage (Section 3.1.1) to guide correct value selection. Once all arguments are filled, the API call is executed to obtain $o_{k_\ell}$. Alongside each API call, we generate a natural-language sub-query $s_{k_\ell}$ that explicitly corresponds to the operation performed by $c_{k_\ell}$. This is constructed based on the API specification, the concrete arguments in API call $c_{k_\ell}$, and the execution context, ensuring that each sub-query reflects both the function being invoked and its role in the overall task flow (Appendix I.5). By iteratively applying this process through the ordered sequence, we construct the full trajectory of call units $\{(s, c, o)\}$, where each element is grounded in real execution and contributes to a coherent, goal-oriented task. This process is depicted in Figure 3b.

**User Query and Response Generation**  Figure 3c depicts the final stage, where once the full set of call units $\{(s, c, o)\}$ has been constructed, we generate a goal-oriented user query $u$ that encapsulates the overall task. The query is created by summarizing the sub-queries $\{s\}$ and abstracting the high-level objective that the entire API workflow is designed to achieve, as detailed in Appendix I.6. Finally, we generate the final response $r$ corresponding to the user query $u$. For this, we provide the LLM with $u$ together with the full set of triplets $(s, c, o)$, as described in Appendix I.7. This allows the model to produce a coherent answer that is grounded in the outputs of the composed API calls.

### 3.1.3 DISCUSSIONS

**Reliable Supervision through Call-First Generation**  Although not targeting goal-oriented tasks, prior studies (Patil et al., 2023; Qin et al., 2023; Tang et al., 2023; Li et al., 2023) have explored using LLMs to construct synthetic data for agent training. They generally follow an **instruction-first strategy** (Wang et al., 2023), where an LLM first generates a user query from API functions, and then produces API calls with arguments corresponding to that query. However, this strategy has intrinsic limitations. Converting a user query into API calls is precisely the capability we aim to improve in LLMs, thus training sets produced by LLMs themselves create a self-reinforcing loop: the model continues to succeed on queries it already handles correctly, but fails to improve on cases where it was originally deficient. This limitation persists even with human verification, as the surviving examples remain biased toward "easy" instances. Consequently, instruction-first generation has primarily been used for distillation, transferring the knowledge of larger closed models into smaller open models. In contrast, we adopt a **call-first strategy**. Here, concrete API calls are instantiated directly from API document, executed to obtain outputs $(c, o)$, and then summarized into a natural language query. This design is advantageous because generating a query from a complete API call is substantially easier for an LLM—requiring only abstraction over given inputs and outputs—than the inverse task of inferring API calls from natural language. By leveraging this easier direction of inference, we transform the summarization and abstraction strengths of LLMs into reliable supervision signals for the reverse direction: learning to map user queries into executable API workflows.

**Benchmark Construction for Goal-Oriented Tasks.**  Existing benchmarks for goal-oriented tasks, such as RestBench (Song et al., 2023) and API-Bank (Li et al., 2023), are built by human annotators who manually identify API relationships and construct corresponding queries and call paths. This costly process severely limits scalability and coverage, leaving existing benchmarks small in size and restricted to evaluation only. In contrast, GOAT automatically captures dependencies across APIs and constructs goal-oriented data in a fully annotation-free manner. This enables cost-effective collection of large-scale data suitable for training. Moreover, with GOAT, evaluation benchmarks with new API sets can be built efficiently using human verification, which is substantially cheaper than full annotation. We further compared a subset of GOAT-generated samples (after human verification) with fully human-annotated data from Song et al. (2023) and Li et al. (2023), and found no substantial difference (examples are in Appendix C). Following this, we built **GOAT-Bench** using the StableToolBench environment (Guo et al., 2024) with APIs from RapidAPI. Details of the benchmark dataset construction and additional examples are provided in Appendix D.

### 3.2 AGENT TRAINING

A central contribution of GOAT is that it turns a setting without any human-annotated training data into a learnable setup by automatically constructing synthetic supervision. On top of this dataset,

we train an LLM agent consisting of a language model and an API retriever. The language model is instruction-tuned with supervision derived from the generated goal-oriented API execution data. Specifically, the model learns to plan the correct sequence of calls and fill in their arguments, and it also receives supervision for generating the final natural-language response by aggregating the execution outputs. To improve generalization beyond specific argument patterns, argument values are masked during training. The retriever is trained using ground-truth query–API document pairs, enabling it to map user queries to the relevant API specifications. Together, this framework enables effective learning of LLM agents tailored for domain-specific API tasks, where both planning and execution are grounded in real API behavior.

## 4 EXPERIMENTS

### 4.1 EXPERIMENT SETUP

We evaluate LLM Agents on goal-oriented API execution benchmarks where no human-annotated training data is available. For each benchmarks, we use Llama-3-70B-Instruct (AI@Meta, 2024) to construct the corresponding synthetic training data within GOAT. While filtering via embedding similarity, we set the hyperparameter $\tau$ to 0.2 for GOATBench and APIBank, while a more conservative threshold of 0.05 is adopted for RestBench. While training, the LLM is fine-tuned with Low-Rank Adaptation (LoRA) method (Hu et al., 2021) to enable parameter-efficient fine-tuning on API execution tasks. For the retriever, we fine-tune a dense retrieval model based on the SBERT architecture (Reimers & Gurevych, 2019), specifically using the all-MiniLM-L6-v2 encoder using InfoNCE loss. All hyperparameters and implementation details are provided in Appendix B.

As a *Baseline* for prompting agents, we followed the decomposition-first method from prior work (Huang et al., 2024); where given a user query, the agent retrieves the top-$k = 5$ relevant API documents and predicts the entire sequence of API calls in a single step. The planned sequence is executed iteratively, with each call incorporating outputs of previous ones, and the final answer is composed from the collected results. This baseline was the strongest among those we tested and was thus chosen as our main baseline, with additional results for the others provided in Appendix E.

### 4.2 EVALUATION BENCHMARKS

**RestBench** (Song et al., 2023)    RestBench is a human-generated benchmark that consists of two test sets built on real-world APIs from TMDB (movie database) and Spotify (music streaming). The evaluation is based on three metrics: (i) *Success%*, measured by human evaluation to assess whether the model result fulfills the user's request; (ii) *Correct Path%*, which counts a case as correct if the gold API call path is contained as a subsequence within the model-generated path, where only the sequence of API functions matters and parameter values are disregarded; and (iii) $\Delta$ *Solution Length*, defined as the mean number of additional API calls inferred, measured only over successful cases, thereby reflecting the efficiency of the generated plan.

**API-Bank** (Li et al., 2023)    API-Bank is constructed with human-implemented API functions in Python, where both the queries and their corresponding API call paths are manually annotated. It consists of three task sets, among which we focus on the *Plan+Retrieve+Call* set, as it uniquely reflects the goal-oriented query setting involving multi-API and multi-call reasoning. The official evaluation metrics include (i) *Correctness*, measured by the precision of API call responses, and (ii) *ROUGE*, computed between the model's final response and the gold response. Since these metrics do not directly capture overall task success, we additionally report *Success%* and *Correct Path%*, following the definitions in RestBench.

**GOATBench**    GOATBench is a human-verified benchmark on real APIs from RapidAPI, constructed through our data generation pipeline with additional human curation (see Appendix D for details). Tasks are categorized into *Single Tool* and *Inter Tool*, depending on whether multiple APIs come from the same tool or from different tools. For evaluation, we adopt three commonly used metrics: (i) *API Selection Accuracy (SA)*, measuring the Jaccard similarity between the predicted and ground-truth sets of API functions (Wang et al., 2024a); (ii) *API Invocation Accuracy (IA)*, similar to SA but requiring all arguments to match in addition to the function name; and (iii) *Success Rate (SR)*, a GPT-based evaluation metric that determines whether the final answer sufficiently and

Table 2: **Experiment results on RestBench.** Closed-source and RestGPT results are reported numbers from original paper, shown here for reference. Metrics are Success%, Correct Path%, and Δ Solution Length. For Vicuna-13B, we additionally reproduced RestGPT using the released code and found substantially lower performance than reported (marked with * in the table).

| | Backbone | Prompting Method | GOAT FT | TMDB Success% ↑ | CP% ↑ | Δ Len. ↓ | Spotify Success% ↑ | CP% ↑ | Δ Len. ↓ |
|---|---|---|---|---|---|---|---|---|---|
| Closed-source | text-davinci-003 | Baseline | - | 29.0 | 33.0 | +1.52 | 14.5 | 36.4 | +1.10 |
| | text-davinci-003 | DEPS | - | 38.0 | 43.0 | +1.20 | 19.3 | 43.8 | +1.74 |
| | text-davinci-003 | ReAct | - | 44.0 | 57.0 | +0.76 | 54.5 | 49.1 | +0.31 |
| | text-davinci-003 | Reflexion | - | 52.0 | 59.0 | +1.37 | 59.6 | 61.4 | +1.68 |
| | text-davinci-003 | RestGPT | - | 75.0 | 79.0 | +0.55 | 72.7 | 74.5 | +0.25 |
| Open-source | Llama2-13B | RestGPT | - | 0.0 | 0.0 | - | 0.0 | 0.0 | - |
| | Llama2-13B | Baseline | ✗ | 0.0 | 0.0 | - | 3.5 | 7.0 | +0.00 |
| | Llama2-13B | Baseline | ✓ | **7.0** | **13.0** | +0.71 | **28.1** | **28.1** | +0.44 |
| | Vicuna-13B | RestGPT | - | 9.0 | **15.0** | +1.21 | 12.7 | 20.6 | +1.52 |
| | Vicuna-13B | RestGPT* | - | 1.0 | 0.0 | +0.00 | 0.0 | 0.0 | - |
| | Vicuna-13B | Baseline | ✗ | 0.0 | 0.0 | - | 0.0 | 0.0 | - |
| | Vicuna-13B | Baseline | ✓ | **17.0** | 14.0 | +0.53 | **29.8** | **33.3** | +1.00 |

Table 3: **Experiment results on API-Bank.** Performance of the API-Bank prompting method is reported numbers from original paper. Since the official inference code is unavailable, additional metrics (Success%, Correct Path%) could not be evaluated.

| | Prompting Method | Backbone | FT Method | Success% ↑ | CP% ↑ | Correctness% ↑ | ROUGE |
|---|---|---|---|---|---|---|---|
| Closed-source | API-Bank | GPT-3 Davinci | - | - | - | 0.00 | 0.0156 |
| | API-Bank | GPT-3.5-turbo | - | - | - | 22.00 | 0.3809 |
| | API-Bank | GPT-4 | - | - | - | 70.00 | 0.4808 |
| Open-source | API-Bank | Alpaca-7B | - | - | - | 0.00 | 0.0860 |
| | API-Bank | ChatGLM-6B | - | - | - | 0.00 | 0.1522 |
| | API-Bank | Llama-7B | API-Bank | - | - | 20.00 | 0.3425 |
| | Baseline | Llama-7B | - | 0.0 | 0.0 | 0.00 | 0.0048 |
| | Baseline | Llama-7B | **GOAT** | **38.0** | **42.0** | **42.22** | 0.3173 |

correctly solves the user query given the tool execution results (Qin et al., 2023). For SR evaluation, we employ GPT-4.1 and provide the exact evaluation prompt in Figure 21.

### 4.3 RESULTS

**Results on RestBench**    Table 2 reports experimental results on RestBench. Without fine-tuning, closed-source models significantly outperform open-source models, reflecting the limited reasoning capabilities of the latter. In fact, baseline results with open-source models are nearly zero in most cases, indicating complete failure. This underscores the necessity of fine-tuning when building on-premise agents with open-source models. Our GOAT training strategy yields clear and consistent improvements over these baselines. Both Llama2-13B and Vicuna-13B achieve substantial gains after fine-tuning, demonstrating the effectiveness of our framework. Remarkably, the fine-tuned models not only surpass the previously reported open-source state-of-the-art method, RestGPT, but in some cases even outperform the closed-source model text-davinci-003 from OpenAI.

**Results on API-Bank**    Table 3 further demonstrates the effectiveness of GOAT through consistent gains on API-Bank. Fine-tuning with GOAT improves performance over zero-shot Llama-7B and, in some cases, even surpasses closed-source models. Compared to the original API-Bank training setup (Li et al., 2023)—which constructs synthetic API functions to train API call usage and relies on instruction-first data generation for multi-instruction queries—our method achieves substantially stronger results by aligning supervision more directly with the target task. While our training method yields improvements across most evaluation metrics, ROUGE is slightly lower. However, as shown in Figure 11, ROUGE is relatively less informative in this context, since LLMs may hallucinate lexically fluent responses that nevertheless bypass correct API execution.

**Results on GOATBench**    As shown in Table 4, GOAT training achieves substantially higher performance compared to models without training. We evaluate its effectiveness with an additional prompting method, ReACT (Yao et al., 2023), and find that GOAT consistently improves performance under both ReACT and the baseline prompting setup. Notably, the gains hold not only over

Table 4: **Experiment results on GOATBench.** Metrics are SA (Selection Accuracy), IA (Invocation Accuracy), and SR (Success Rate).

| | Prompting Method | Backbone | FT Method | Inter Tool | | | Single Tool | | |
|---|---|---|---|---|---|---|---|---|---|
| | | | | SA | IA | SR | SA | IA | SR |
| Closed-source | Baseline | GPT-4.1 | - | 22.7 | 13.7 | 40.8 | 27.9 | 18.8 | 54.3 |
| Open-source | ReACT | Llama2-7B | - | 1.9 | 0.2 | 1.8 | 2.6 | 0.0 | 5.2 |
| | ReACT | Llama2-7B | ToolLLM | 12.5 | 1.0 | 3.2 | 26.8 | 0.5 | 2.6 |
| | ReACT | Llama2-7B | **GOAT** | **24.6** | **4.6** | **3.7** | **41.8** | **7.6** | **6.4** |
| | Baseline | Llama3-8B | - | 10.4 | 3.3 | 4.1 | 18.6 | 6.0 | 7.1 |
| | Baseline | Llama3-8B | **GOAT** | **59.0** | **26.1** | **14.4** | **68.9** | **35.6** | **24.5** |

Table 5: **Comparison across backbone models on GOATBench.** We used *Baseline* as prompting method.

| Backbone | FT | Inter Tool | | | Single Tool | | |
|---|---|---|---|---|---|---|---|
| | | SA | IA | SR | SA | IA | SR |
| Qwen2-7B | ✗ | 18.9 | 6.2 | 2.5 | 29.0 | 6.1 | 6.4 |
| Qwen2-7B | ✓ | **39.8** | **15.8** | **3.0** | **46.2** | **14.1** | **7.2** |
| Llama3-8B | ✗ | 10.4 | 3.3 | 4.1 | 18.6 | 6.0 | 7.1 |
| Llama3-8B | ✓ | **59.0** | **26.1** | **14.4** | **68.9** | **35.6** | **24.5** |
| Llama3-70B | ✗ | 15.1 | 5.0 | 14.3 | 33.2 | 10.3 | 19.4 |
| Llama3-70B | ✓ | **55.7** | **18.9** | **20.2** | **62.8** | **18.3** | **39.0** |

Table 6: **Retriever performance on GOATBench.** We use all-MiniLM-L6-v2 encoder as a retriever. Recall@GT is computed by retrieving the same number of documents as the ground-truth API calls for each data point.

| GOAT FT | Recall@GT | Recall@5 |
|---|---|---|
| ✗ | 25.4 | 41.9 |
| ✓ | **63.3** | **83.4** |

the non-fine-tuned model but also over the fine-tuned one on the instruction-first data generated by ToolLLM (Qin et al., 2023), operating with ReACT. Although ToolLLM also generates synthetic data using the same APIs as GOATBench, its data construction process is not tailored to goal-oriented queries but instead emphasizes parallel multi-API queries with fine-grained instructions, resulting in poor performance on the goal-oriented evaluation.

**Effects of GOAT with Diverse LLMs**   Table 5 further confirms that GOAT consistently boosts performance across different LLM backbones, including Qwen2-7B, Llama3-8B, and Llama3-70B. These results highlight the robustness and generality of our fine-tuning strategy. Notably, even for Llama3-70B, where the same model was used both to generate training data and to perform fine-tuning, we still observe substantial gains. As discussed in Section 3.1.3, this is because our call-first generation strategy provides reliable supervision: by starting from executable API paths and asking the model only to abstract them into natural-language queries, we avoid pitfalls of instruction-first methods and enable the model to benefit from supervision signals even when self-generated.

**Results of Document API Retriever**   We conduct an ablation study to evaluate the retriever performance in isolation using recall metrics on GOATBench. Table 6 reports Recall@GT and Recall@5 when retrieving relevant API documents given the user query $u$. Recall measures how many of the retrieved documents are actually correct. Recall@GT evaluates this by retrieving the same number of documents as the ground-truth API calls for each data point. The results show clear performance improvements through fine-tuning.

## 5   CONCLUSIONS

In this work, we present **GOAT**, a fully automatic training framework for equipping LLM agents with goal-oriented reasoning capabilities over interdependent APIs. Unlike prior approaches that rely solely on zero-shot evaluation in the absence of training data, GOAT exploits existing API documentation to automatically construct synthetic supervision. This allows open-source LLMs to be fine-tuned efficiently while retaining strong generalization across diverse goal-oriented scenarios. Furthermore, leveraging GOAT pipeline, we curated **GOATBench**, a benchmark for evaluating goal-oriented tool use. Extensive experiments on GOATBench and other benchmarks demonstrate that GOAT-trained agents not only achieve state-of-the-art performance among open-source models but also, in some cases, surpass closed-source systems with strong reasoning ability. Overall, our results highlight GOAT as a practical and scalable path toward building robust open-source LLM agents that can reason over complex tool interactions and deliver accurate responses to high-level user goals.

ETHICS STATEMENT

We acknowledge the importance of transparency and responsible use of publicly available resources in our research. All datasets, benchmarks, models, and APIs employed in this study were publicly released for academic research and were used in full compliance with their respective licenses and intended purposes. Specifically, we used the Meta-Llama-3-8B-Instruct model under the Llama 3 Community License, the Sentence-BERT model under the Apache 2.0 License, and publicly available benchmarks such as RestGPT and StableToolBench, as well as the TMDB API under its non-commercial usage terms. No proprietary or private data were used and this research does not involve human subjects or sensitive personal information.

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

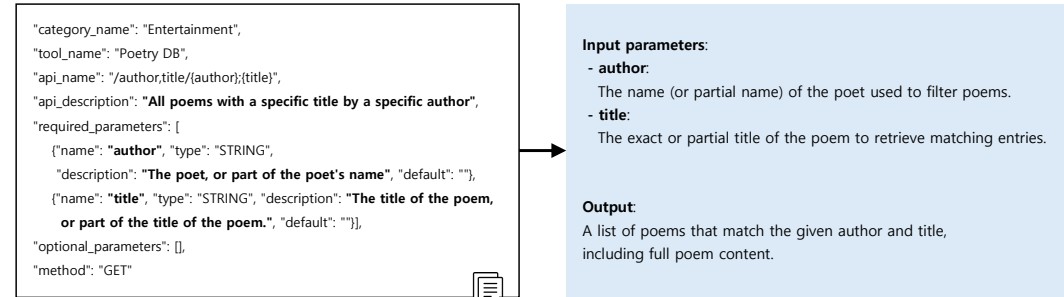

Figure 4: **Example of API document parsing result.**

Table 7: **Precision and recall of each pruning stage.**

|  | Embedding Similarity | LLM | API Call Execution |
|---|---|---|---|
| Precision | 0.25 | 0.59 | 0.90 |
| Recall | 0.92 | 0.42 | 0.36 |

# A  API DEPENDENCY GRAPH CONSTRUCTION

## A.1  API DOCUMENT PARSING EXAMPLE

Figure 4 illustrates an example of how raw API document is converted into structured representations. By prompting LLM as in Figure 13, we extract both the types and number of input parameters of each API function together with their semantic roles, as well as the semantic meaning of the returned output.

## A.2  FILTERING VALID DEPENDENCY EDGES

GOAT adapt three-step filtering process to obtain valid API dependency edges. The precision and recall of each pruning stage are summarized in Table 7. The first row represents the precision, while the second row corresponds to the recall, based on 500 data edges that we manually annotated. The results demonstrate that our method successfully identifies valid edges $e = (n_i, n_j, k)$ while filtering out spurious connections, thereby enhancing the quality of the constructed graph.

## A.3  API DEPENDENCY GRAPH EXAMPLE

Figure 5 shows example of constructed API dependency graph.

# B  IMPLEMENTATION DETAILS

Fine-tuning of LLM model was performed on a single NVIDIA H100 GPU for 3 epochs on every experiment. We adopted the Low-Rank Adaptation (LoRA) method Hu et al. (2021) with $r = 8$, $\alpha = 16$, and a dropout ratio of 0.05 to enable parameter-efficient fine-tuning on instruction-guided API execution tasks. All experiment results reported in this paper are based on a single run without variation across random seeds.

We additionally provide the data statistics used in each training stage. API calls occur during dependency-graph construction (roughly two calls per edge) and during synthetic data generation (one call per node), and all endpoints used in our experiments (TMDB, Spotify, and StableTool-Bench APIs) are free services. The number of synthetic instances used for LLM fine-tuning is 8570 for RestGPT–TMDB, 924 for RestGPT–Spotify, 108 for API-Bank, and 1631, 1354, 650, and 420 for the Entertainment, Financial, Food, and Travel subsets of GOATBench. The corresponding numbers of query–document pairs used for SBERT training are 33169 for RestGPT–TMDB, 3389 for RestGPT–Spotify, 180 for API-Bank, and 5091, 4752, 1957, and 1166 for the four GOATBench domains.

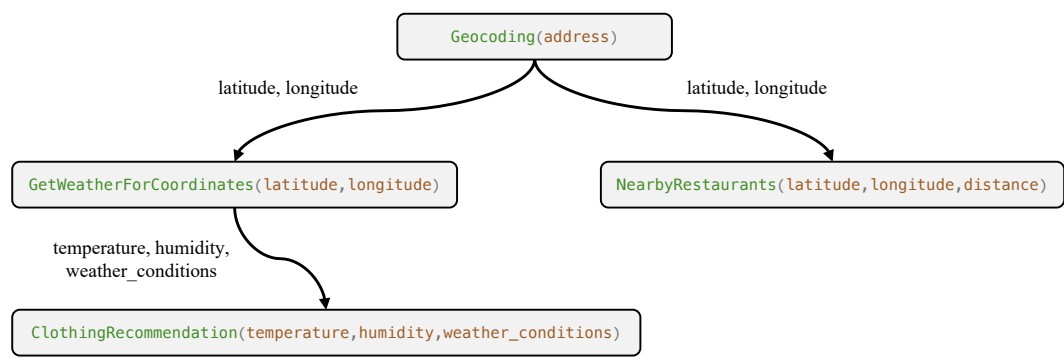

Figure 5: **Example of Constructed API Dependency Graph from APIBank APIs.**

Table 8: Statistical comparison of GOATBench, RestBench, and API-Bank.

|  | **GOATBench** | **RestBench** | **API-Bank** |
|---|---|---|---|
| # of Domains | 4 | 2 | 8 |
| # of APIs | 182 | 94 | 73 |
| # of Instances | 747 | 157 | 314 |
| Avg. Len | 3.1 | 2.41 | 2.91 |

## C   GOAT-GENERATED DATA QUALITY ANALYSIS

As we discussed in Section 3.1.3, GOAT enables benchmark curation with much less human effort than other fully human annotated benchmarks such as RestBench Song et al. (2023) and API-Bank Li et al. (2023). Figures 6 to 8 show qualitative comparsion between GOAT-generated data and fully human-generated data examples.

## D   GOATBENCH

**GOATBench** is a human-verified benchmark built on top of the GOAT framework. It consists of 747 goal-oriented API execution tasks, where solving each task requires planning and invoking a sequence of interconnected APIs. Among them, 372 tasks belong to the *seen* category and 375 to the *unseen* category, enabling evaluation across both familiar and novel API compositions. While the *seen* set serves as the primary evaluation target, the *unseen* set provides an additional measure of generalization to previously unseen APIs. This benchmark was constructed with the GOAT pipeline and subsequently verified and annotated by human experts. Qualitative examples are shown in Figure 9.

The benchmark covers 291 tools and 182 APIs across four user-centric domains—*financial*, *food*, *entertainment*, and *travel*—with all APIs collected from RapidAPI Hub via StableToolBench Qin et al. (2023). RapidAPI provides a hierarchical structure where each tool contains multiple APIs. In addition, we implemented two global tools—*compare* and *difference*—that apply across all domains. The *compare* tool evaluates numerical similarity or proximity between values, while the *difference* tool determines which value is greater, enabling comparison-based reasoning. All LLM-based construction stages use the LLaMA-3-70B-Instruct model AI@Meta (2024), ensuring reproducibility and easy extension to other API sets.

The *seen test* set is created by sampling subgraphs from the same API dependency graphs as training, while modifying parameter values to ensure semantic variation under identical structures and call sequences. This set is the main benchmark for measuring performance in in-domain settings. The *unseen test* set includes tasks involving tools absent from training, serving as a secondary evaluation to test generalization. Following Qin et al. (2023), we also categorize tasks into two types: **Single Tool**, where multiple APIs under the same tool (i.e., a collection of related APIs from a single RapidAPI service) are composed to solve the task; and **Inter Tool**, which requires composing APIs

Table 9: Distribution of subgraph sizes in GOATBench.

| Subgraph Size | # of Instances |
|---|---|
| 2 | 90 |
| 3 | 480 |
| 4 | 163 |

Table 10: **Experiment results on seen test set.** ID: Instruction Decomposer. **GOAT FT**: GOAT fine-tuned. We use Llama-3-8B-Instruct as backbone model. Metrics are SA (Selection Accuracy), IA (Invocation Accuracy), and SR (Success Rate).

| Prompting Method | GOAT FT | Inter Tool | | | Single Tool | | |
|---|---|---|---|---|---|---|---|
| | | SA | IA | SR | SA | IA | SR |
| ReACT | ✗ | 13.3 | 2.6 | 9.2 | 20.7 | 1.9 | 9.7 |
| | ✓ | **51.9** | **20.3** | **13.4** | **57.6** | **30.9** | **18.0** |
| ReACT + ID | ✗ | 15.2 | 3.4 | 6.9 | 23.5 | 7.2 | 5.8 |
| | ✓ | **37.1** | **14.9** | **8.3** | **42.6** | **30.7** | **8.4** |
| Global (**Baseline**) | ✗ | 10.4 | 3.3 | 4.1 | 18.6 | 6.0 | 7.1 |
| | ✓ | **59.0** | **26.1** | **14.4** | **68.9** | **35.6** | **24.5** |
| Global + ID | ✗ | 19.0 | 11.9 | 9.2 | 35.6 | 26.0 | **12.9** |
| | ✓ | **53.7** | **25.0** | **10.6** | **67.6** | **33.2** | 11.6 |

across different tools, testing the agent's ability to reason over heterogeneous services and handle tool chaining.

A detailed statistical summary of GOATBench, RestBench, and API-Bank is provided in Table 8, including the number of domains, APIs, instances, and average dependency-chain lengths. This comparison highlights the complexity and coverage of GOATBench relative to prior benchmarks. Table 9 further reports the distribution of subgraph sizes used to construct GOATBench tasks, illustrating the prevalence of 2–4 step dependency chains in real API compositions.

# E  LLM AGENT PROMPTING BASELINES

We test our approach using following four LLM agent prompting baaselines designed to perform goal-oriented tasks. Each LLM agent design consists of four key components: API retrieval, API selection, API call generation, and final response generation. The agent designs differ in how and at what level the LLM performs planning.

**ReACT** (Yao et al., 2023)  ReACT is a reflective agent that performs planning in an iterative manner. Since a goal-oriented user query often requires a sequence of dependent API calls, ReACT can serve as the baseline planning method. At each timestep, it jointly selects the next API to call and generates its arguments, conditioned on the full history of prior API calls and their outputs. Given the user query $u$, $k$ potentially relevant documents are first retrieved using an external retriever. The same set of the retrieved documents are then fed to the agent at each timestep.

**Global Planner** (Huang et al., 2024)  Given $k$ relevant documents retrieved as in ReACT, this agent performs global planning by determining the entire sequence of API functions with a single prompt, using only the user query and the retrieved API documents. This global planning strategy enables the agent to optimize the overall call sequence from a holistic perspective. To reflect the outputs of the previous API calls, we iteratively generate the API calls based on the planned API function sequence. Then, API calls are executed iteratively, incorporating the previously executed API outputs.

**ReACT + Instruction Decomposer**  This agent extends ReACT by incorporating iterative planning at the natural language level. Starting from the user query, it generates one subinstruction at a time, conditioned on the original query, prior subinstructions, API calls, and their outputs. For each subinstruction, it retrieves the $k$ most relevant API document and jointly performs API selection

Table 11: **Experiment results on unseen test set (Global agent).** Full performance table is in Appendix G.

| Prompting Method | GOAT FT | Inter Tool | | | Single Tool | | |
|---|---|---|---|---|---|---|---|
| | | SA | IA | SR | SA | IA | SR |
| Baseline | ✗ | 11.1 | 5.2 | 11.5 | 19.5 | 9.1 | 27.1 |
| Baseline | ✓ | **42.0** | **16.7** | **19.4** | **37.2** | **26.1** | **34.6** |

Table 12: **Ablation results on different LLM fine-tuning methods with Global design.** We use Llama-3-8B-Instruct model as backbone. All models use finetuned SBERT as the retriever.

| LLM FT Method | Inter Tool | | | Single Tool | | |
|---|---|---|---|---|---|---|
| | SA | IA | SR | SA | IA | SR |
| LoRA | 58.9 | 24.5 | 11.6 | **69.2** | 34.2 | 17.4 |
| LoRA + Self-Distill | 42.7 | 17.2 | 6.0 | 53.7 | 31.9 | 10.3 |
| LoRA + Masking Args | **59.0** | **26.1** | **14.4** | 68.9 | **35.6** | **24.5** |

and call generation. This process continues step-by-step, with each execution result guiding the generation of the next subinstruction.

**Global Planner + Instruction Decomposer** This agent also performs language-based subinstruction planning but returns global plans in a single shot. For each subinstruction, it then retrieves $k$ relevant API documents, and jointly performs API selection and call generation in the given subinstruction order, reflecting the planned execution flow.

Table 10 compares the results of applying GOAT fine-tuning across different prompting methods. We observe consistent performance improvements over all methods, demonstrating the broad effectiveness of our approach. Among them, the decomposition-first method with a global planning strategy achieves the largest gains, and we therefore adopt it as our primary baseline for the main experiments.

## F  QUALITATIVE RESULTS

Qualitative result examples on each benchmark are in Figures 10 to 12.

## G  RESULTS ON UNSEEN TEST

Although the unseen setting is not the primary target of our benchmark, we observe that models trained on GOAT still exhibit strong generalization performance. As shown in Table 11, fine-tuning both the LLM and retriever leads to consistent improvements across all task types. This demonstrates that task-aligned fine-tuning not only benefits in-domain execution but also improves robustness to previously unseen API combinations, though the gains are more limited compared to the seen setting.

## H  ABLATION RESULTS ON LLM FINE-TUNING METHODS

Table 12 compares different fine-tuning strategies designed to mitigate overfitting in the LLM. We observed that standard LoRA-based fine-tuning tends to cause the model to overfit to specific argument values seen during training, leading to reduced generalization. To address this issue, we explored two approaches: (1) self-distillation with soft targets from a pretrained model, and (2) masking argument tokens by setting their loss contributions to zero. Our results show that the masking strategy is particularly effective, as it prevents memorization of argument values and encourages structural learning of API call formats.

## I DATA GENERATION PROMPTS

### I.1 API DOCUMENT PARSING PROMPT

See Figure 13 for an example prompt.

### I.2 LLM FILTERING PROMPT

See Figure 14 for an example prompt.

### I.3 ACTUAL CALL OUTPUT FILTERING PROMPTS

See Figure 15, Figure 16 for an example prompt.

### I.4 API CALL SEQUENCE GENERATION PROMPT

See Figure 17 for an example prompt.

### I.5 SUB-INSTRUCTION GENERATION PROMPT

See Figure 18 for an example prompt.

### I.6 USER QUERY GENERATION

See Figure 19 for an example prompt.

### I.7 FINAL RESPONSE GENERATION

See Figure 20 for an example prompt.

## J SUCCESS RATE PROMPT

See Figure 21 for prompt used in SR evaluation on GOATBench.

## K USE OF LLMS

We acknowledge that LLMs were used as writing assistants to improve grammar, clarity, and readability of the manuscript.

GOAT-generated Example

```
{
    "query": "Get the keywords for movie 'The Shawshank Redemption' released in 1994.",
    "api_path": [
        {
            "api_name": "GET /search/movie",
            "input": {
                "query": "The Shawshank Redemption",
                "include_adult": false,
                "region": "US",
                "year": 1994,
                ...
            },
            "output": {
                "error": "",
                "response": {
                    "page": 1,
                    "results": [
                        {
                            ...
                            "id": 278,
                            "original_title": "The Shawshank Redemption",
                            ...
                        }
                    ],
                    ...
                }
            },
            "sub_instruction": "Search for movies with the query
            'The Shawshank Redemption' in the US region, released in 1994,
            on the first page, excluding adult content."
        },
        {
            "api_name": "GET /movie/{movie_id}/keywords",
            "input": {
                "movie_id": 278
            },
            "output": {
                "error": "",
                "response": {
                    "id": 278,
                    "keywords": [
                        {"id": 378, "name": "prison"},
                        {"id": 6054, "name": "friendship"},
                        {"id": 796, "name": "police brutality"},
                        {"id": 417, "name": "corruption"},
                        {"id": 818, "name": "based on novel or book"},
                        {"id": 6089, "name": "freedom"},
                        ...
                    ]
                }
            },
            "sub_instruction": "Retrieve the keywords associated with the movie
            'The Shawshank Redemption' which has an ID of 278."
        }
    ],
    "final_response": "The keywords for the movie 'The Shawshank Redemption' released in
    1994 are: prison, friendship, police brutality, corruption, based on novel or book,
    freedom, and more."
}
```

Human-generated Example

```
{
    "query": "What are some common keywords associated with movie \"Titanic\"?",
    "solution": [
        "GET /search/movie",
        "GET /movie/{movie_id}/keywords"
    ]
}
```

Figure 6: **Comparison of Goat-Generated and Human-generated on TMDB.**

**GOAT-generated Example**

```
{
    "query": "Get me an album of the first artist I'm following.",
    "api_path": [
        {
          "api_name": "GET /me/following",
          "input": {
            "type": "artist",
            "after": "",
            "limit": 20
          },
          "output": [
            {
                ...
                "total": 3,
                "items": [
                  {
                    "id": "74ASZWbe4lXaubB36ztrGX",
                    "name": "Bob Dylan",
                    ...
            ]
              "sub_instruction": "Retrieve the current user's followed artists, with a
              maximum of 20 results, using the 'artist' ID type and starting from the
              beginning."
        },
        {
          "api_name": "GET /artists/{id}/albums",
          "input": {
            "id": "74ASZWbe4lXaubB36ztrGX",
            ...
          },
          "output": [
            {
                ...
                "items": [
                  {
                    "id": "6y2WHyqRUCeHrjMXvjnRmD",
                    "name": "Modern Times",
                    "album_type": "album",
                    "total_tracks": 20,
                    "release_date": "2006-08-29",
                    ...,
                    "artists": [
                      {
                        "id": "74ASZWbe4lXaubB36ztrGX",
                        "name": "Bob Dylan",
                        "external_urls": {
                          "spotify": "https://open.spotify.com/artist/74ASZWbe4lXaubB36..."
                        }
                    ...
            ]
              "sub_instruction": "Retrieve the album information for the artist with id
              '74ASZWbe4lXaubB36ztrGX', which is the first artist in the list of followed
              artists, with a limit of 20 items and an offset of 0, without any specific
              market or album type filter."
        }
    ],
    "final_response": "The first artist you are following is Bob Dylan. One of his albums
    is 'Modern Times' (released on August 29, 2006), which contains 20 tracks. You can
    explore more of his albums on Spotify."
}
```

**Human-generated Example**

```
{
    "query": "Recommend more artists base on my first following artist",
    "solution": [
        "GET /me/following",
        "GET /artists/{id}/related-artists"
    ]
}
```

Figure 7: **Comparison of Goat-generated and Human-Generated on Spotify.**

**GOAT-generated Example**

```
{
    "query": "What are the tax deductions for a Data Scientist's salary?",
    "api_path": [
        {
            "api_name": "GetOccupationSalary",
            "input": {
                "occupation": "Data Scientist"
            },
            "output": "{\"salary\": 150000}",
            "sub_instruction": "Retrieve the salary for the occupation 'Data Scientist'."
        },
        {
            "api_name": "TaxCalculator",
            "input": {
                "salary": 150000.0
            },
            "output": "{\"salary_after_tax\": 105000.0}",
            "sub_instruction": "Calculate the tax deductions for a salary of 150000.0."
        }
    ],
    "final_response": "As a Data Scientist, your take-home salary after tax deductions
    would be around $105,000, based on a gross salary of $150,000."
}
```

**Human-generated Example**

```
{
    "query": "Calculate the monthly salary after taxes for a Data Scientist",
    "api_path": [
        {
            "api_name": "GetOccupationSalary",
            "input": {
                "occupation": "Data Scientist"
            },
            "output": {
                "api_name": "GetOccupationSalary",
                "input": {
                    "occupation": "Data Scientist"
                },
                "output": {
                    "salary": 150000
                },
                "exception": null
            }
        },
        {
            "api_name": "TaxCalculator",
            "input": {
                "salary": "150000"
            },
            "output": {
                "api_name": "TaxCalculator",
                "input": {
                    "salary": 150000.0
                },
                "output": {
                    "salary_after_tax": 105000.0
                },
                "exception": null
            }
        }
    ],
    "final_response": "For a Data Scientist with a salary of $150,000, the monthly salary
    after taxes is $105,000.",
}
```

Figure 8: **Comparison of Goat-generated and Human-Generated on APIBank.**

1080
1081
1082
1083
1084
1085
1086
1087
1088
1089
1090
1091
1092
1093
1094
1095
1096
1097
1098
1099
1100
1101
1102
1103
1104
1105
1106
1107
1108
1109
1110
1111
1112
1113
1114
1115
1116
1117
1118
1119
1120
1121
1122
1123
1124
1125
1126
1127
1128
1129
1130
1131
1132
1133

**GOATBench Data Example**

```json
{
    "query": "Can you help me find a hotel near the Flixbus stations in Paris?",
    "api_path": [
        {
            "api_name": "Flixbus_v2/Autocomplete",
            "input": {
                "query":"Paris",
                "locale":"en"
            },
            "output": {[
                {
                "id": "dcc05c30-9603-11e6-9066-549f350fcb0c",
                "name": "Paris CDG Airport",
                "address": "Gare Routière Roissypole",
                ...
                },
                {
                "id": "7f6214cd-95d7-4bbf-8283-55af226fa33a",
                "name": "Paris (Pont de Sèvres)",
                ...
                },
                ...
            ]},
            "subinstruction": "Search for Flixbus stations using the query 'Paris'
            and retrieve results in English."
        },
        {
            "api_name": "Hotels/locations/v2/search",
            "intput": {
                "query":"Paris CDG Airport",
                "locale":"en_US",
                "currency":"EUR"
            },
            "output": {[
                {
                "name": "Paris Marriott Charles de Gaulle Airport Hotel",
                "geoId": "5854",
                ...
                },
                {
                "name": "Holiday Inn Express Paris – CDG Airport, an IHG Hotel",
                "geoId": "19497017",
                ...
                },
                ...
            ]},
            "subinstruction": "Find hotels near Paris CDG Airport station."
        }
    ],
    "final_response": "There is a Flixbus stop at Paris CDG Airport (Gare Routière
    Roissypole). You can book nearby hotels such as the Paris Marriott Charles
    de Gaulle Airport Hotel or Holiday Inn Express Paris – CDG Airport."
}
```

Figure 9: **Qualitative Example of GOATBench data.**

**Zero-shot Inference Example**

```
{
    "user query": "What is the birthday of the most popular person right now?",
    "api_path": [],
    "final_response": "I apologize, but the API did not provide any data on the most
    popular person. Please provide a valid API response or clarify the query to get
    a more accurate answer."
}
```

**GOAT Fine-tuned Inference Example**

```
 {
    "user query": "What is the birthday of the most popular person right now?",
    "api_calls": [
        {
            "api_name": "GET /person/popular",
            "input": {
                "page": 1
            },
            "output": {
              "id": 53,
              "name": "Peter Stormare",
              "original_name": "Peter Stormare",
              ...
            },
        {
            "api_name": GET /person/{person_id},
            "input": {
                "person_id": "53"
            },
            "output": {
              "id": 53,
              "name": "Peter Stormare",
              "birthday": "1953-08-27",
              ...
            }

        }
    ],
    "final_response": "The most popular person is Peter Stormare, a Swedish actor
    born on August 27.",
}
```

Figure 10: **Comparison of zero-shot inference result and GOAT fine-tuned inference result on RestBench.**

```
Zero-shot Inference Example

{
    query": "Retrieve a list of recommended movies based on Peter's preferences
    and filter out the movies that the user has already watched.",
    "api_path": [
        {
            "api_name": "UserMoviePreferences",
            "input": {
                "user_name": "Peter"
            },
            "output": {
                "preferences": ["Action", "Drama","Thriller"]
            }
        },
        {
            "api_name": "UserWatchedMovies",
            "input": {
                "user_name": "Peter"
            },
            "output": ["The Matrix", "The Lord of the Rings","The Dark Knight"]
        },
        {
            "api_name": "MovieRecommendations",
            "input": {},
            "output": {"error": "missing 1 required positional argument: 'preferences'"}
        }
        ],
        "final_response": "Based on Peter's preferences for Action, Drama, and Thriller,
        here are some recommended movies: The Shawshank Redemption,
        The Silence of the Lambs, The Dark Knight Rises."
}
```

```
GOAT Fine-tuned Inference Example

{
    "query": "Retrieve a list of recommended movies based on Peter's preferences
    and filter out the movies that the user has already watched.",
    "api_path": [
        {
            "api_name": "UserMoviePreferences",
            "input": {
                "user_name": "Peter"
            },
            "output": {
                "preferences": ["Action", "Drama","Thriller"]
            }
        },
        {
            "api_name": "UserWatchedMovies",
            "input": {
                "user_name": "Peter"
            },
            "output": ["The Matrix", "The Lord of the Rings","The Dark Knight"]
        },
        {
            "api_name": "MovieRecommendations",
            "input": {
                "preferences": ["Action", "Drama","Thriller"]
            },
            "output": {
                "recommended_movies": [
                    "The Dark Knight",
                    "The Lord of the Rings",
                    "The Shawshank Redemption",
                    "Forrest Gump",
                    ...
        ],
        "final_response": "Based on your preferences for Action, Drama, and Thriller movies,
        I recommend the following movies that you haven't watched yet:
        The Shawshank Redemption, Forrest Gump, Joker, Inception, Parasite, and Get Out."
}
```

Figure 11: **Comparison of zero-shot inference result and GOAT fine-tuned inference result on API-Bank.**

**Zero-shot Inference Example**

```
{
    "query": "Get the detailed information from the search results of books with
    the name 'harry potter'.",
    "api_path": [],
    "final_response": "No search results found for 'Harry Potter'."
}
```

**GOAT Fine-tuned Inference Example**

```
{
    "query": "Get the detailed information from the search results of books with
    the name 'harry potter'.",
    "api_path": [
        {
            "api_name": "HAPI Books/Search Books by Name",
            "input": {
                "book_name": "harry+potter"
            },
            "output": [
              {
                "book_id": 136251,
                "name": "Harry Potter and the Deathly Hallows",
                ...
              },
              {
                "book_id": 42844155,
                "name": "Harry Potter and the Sorcerer's Stone",
                ...
              },
              ...
            ]
        },
        {
            "api_name": "HAPI Books/Get Book Information by Book Id",
            "input": {
                "book_Id": "136251"
            },
            "output": {
              "name": "Harry Potter #7Harry Potter and the Deathly Hallows",
              "authors": ["J.K. Rowling"],
              "rating": 4.62,
              "pages": 759,
              "published_date": "July 21, 2007",
              "synopsis": "Harry has been burdened with a seemingly impossible task:
              that of locating and destroying Voldemort's remaining Horcruxes. ..."
            }
        }
    ],
    "final_response": "Here are the results of the search for books with the name
    'Harry Potter': Harry Potter and the Deathly Hallows by J.K. Rowling, published in
    2007, has a rating of 4.62. Here is a summary of Harry Potter and the Deathly Hallows:
    Harry has been tasked with locating and destroying Voldemort's remaining Horcruxes,
    ... "
}
```

Figure 12: **Comparison of zero-shot inference result and GOAT fine-tuned inference result on GOATBench.**

---

### API Document Parsing Prompt

```
You are an API Documentation Assistant responsible for analyzing API documentation
and summarizing the semantics of each input parameter and the output of the API function.

You will be provided with:
1. API Document: A dictionary containing information about an API function, with details.

Your task is to:
1. Provide a clear semantic description of what each input parameter
and output of the API function represents.
2. There can be multiple input parameters, including both required and optional
parameters.
3. If there are no required or optional parameters, return empty array
for input parameter description.

Output Format:
- You must return a dictionary with the keys "input_params" and "output".
- "input_params": Return an array of semantic descriptions for each input parameter.
                  If there is None, return empty array.
- "output": Return a semantic description for output of the API function.

ONLY return the dictionary as your output. DO NOT include any other words.
```

Figure 13: **Prompt used for API document parsing.**

---

### LLM Filtering Prompt

```
You are an API Documentation Assistant responsible for determining whether two APIs
can be connected
sequentially, i.e. the output of the first API must be used as the input for the
second API.

You will be provided with:
1. API1 Document: A dictionary containing the details of API1's output.
2. API1 Semantic Descriptions: Natural language explanations of API1's output
3. API2 Document: A dictionary containing the details of API2's input.
4. API2 Semantic Descriptions: Natural language explanations of API2's input.

Your task is to:
1. Analyze the semantic descriptions and the provided API documents to determine if
API1's output
   can be used as API2's input.
2. Return True only if the information in the output of API1 can be used as a valid
input for API2.
3. Do not return True when input of API1 can be reused in API2.
4. Explain why the APIs are connectable or not.

Output Format:
- You must return a dictionary with the keys "connectable" and "reason".
- "connectable": Return True only if API1's output can be used as API2's input,
otherwise return False.
- "reason": Provide a clear explanation describing why the APIs can or cannot be
connected.

ONLY return the dictionary as your output. DO NOT include any other words.
```

Figure 14: **Prompt used for filtering edges via LLM.**

---

### API Call Generating Prompt for Edge Filtering - 1. First Call

```
You are an API Documentation Assistant responsible for generating function calls
based on API documentation.

You will be provided with:
1. API Document: A dictionary containing information about an API function, with details.

Your task is to:
1. Create a fictional scenario where you need to use the API.
2. Populate the API function's required parameters and optional parameters with
appropriate values, ensuring that all required parameters are included and match the
correct data types.

Output Format:
- You must return a dictionary where each parameter name is the key, and the parameter
  value is the value of the dictionary.
- Ensure each parameter value has the correct data types.
- If there are no required or optional parameters for the API function, return an empty
  dictionary.

ONLY return the parameter dictionary as your output. DO NOT include any other words.
```

### API Call Generating Prompt for Edge Filtering - 2. Subsequent Call

```
You are an API Documentation Assistant responsible for generating function calls
based on API documentation and previous API call results.

You will be provided with:
1. API Document: A dictionary containing information about an API function,
   including parameter names, data types, and descriptions.
2. API Call Results: The result of one or more previous API function calls.
3. Reason: An array explaining how the API Call Results can be used to populate
   the parameters for the current API call.

Your task is to:
1. Create a fictional scenario where you need to use the API.
2. Populate the API function's required and optional parameters using the following rules:
   - First, use values justified by the API Call Results and the Reason array.
   - If a parameter cannot be filled this way, infer it using the information in the API
   Document
     (e.g., parameter descriptions or type hints).
3. Ensure all parameter values match the correct data types as specified in the API
   Document.

Output Format:
- Return a dictionary where each key is a parameter name and the value is the parameter's
  value.
- If no parameters can be populated from the available information, return an empty
  dictionary.

ONLY return the parameter dictionary as your output. DO NOT include any other text.
```

Figure 15: **Prompt used for API call generation for each edges.**

**API Call Filtering Prompt**

```
You are an API Documentation Assistant responsible for determining if the information from
the result of the first API call is used in the parameters of the second API call.
You will be provided with:
1. api_result: A result from the first API call.
2. llm_result: Parameters and their values for calling next API.

Your task is to:
1. Analyze the contents of api_result to determine if it was used as input in llm_result.
2. Provide an explanation about whether or not the first API result influenced the
   parameters of the next API call.

Output Format:
- You must return a dictionary with the keys "connectable" and "reason".
- "connectable": Return True if api_result was used in llm_result, otherwise return False.
- "reason": Provide a clear explanation describing why api_result was or was not used
  as part of llm_result.

ONLY return the dictionary as your output. DO NOT include any other words.
```

Figure 16: **Prompt used for filtering edges via API Call Output.**

**Make First Call**

```
You are an API Documentation Assistant responsible for constructing parameter values
for API calls based on API documentation.
You will be provided with:
1. API Document: A dictionary containing information about an API function, with details.
Your task is to:
1. Create a fictional scenario where you need to use the API.
2. Populate the API function's required parameters and optional parameters
   with appropriate values, ensuring that all required parameters are included
   and match the correct data types.
Output Format:
- Return a dictionary where each parameter name is the key, and the parameter value is
  the value of the dictionary.
- Ensure each parameter value has the correct data types.
- If there are no required or optional parameters for the API function, return an empty
  dictionary.
ONLY return the parameter dictionary as your output. DO NOT include any other words.
```

## Make Call Step 1

You are an API Documentation Assistant responsible for constructing parameter values
for API calls based on API documentation and previous API call results.
You will be provided with a dictionary containing the following keys:
1. `API Document`:
   This key provides information about an API function, including its details.
   It should be used solely to understand the API and identify its required and optional
   parameters.
   - **Important:** Do not use any values from the `API Document` directly to populate
   parameters
   for the API call.
2. `Parameter Dictionary`:
   This key contains a dictionary where each key is a parameter index, and each value is
   the corresponding parameter name. This is used to reference parameters by their
   indices.
3. `Parameter Value`:
   This key contains a dictionary that maps each parameter index to a dictionary detailing
   how to obtain the parameter's value based on previous API call results:
   - Each value includes:
     - `docid`: The unique ID of the document from which the parameter value is derived.
     This `docid` corresponds directly to a `docid` in the `Previous Result`, indicating
     the source of the data to be used.
     - `reason`: A brief explanation of how the specific data from the previous results
     (API1)
     is suitable to be used as a parameter in the current API call (API2).
4. `Previous Result`:
   This key contains a dictionary of results from previous API function calls.
   Each key is a `docid` that corresponds to a previous API call, and each value contains
   the results returned by that call. The `docid` used here matches the `docid` referenced
   in the `Parameter Value`.
### Your task is to follow these steps:
1. **Identify Parameter Names**:
   - Use the `Parameter Dictionary` to reference the names of parameters using their
   indices
   provided in the `Parameter Value`.
2. **Extract Parameter Values**:
   - For each parameter identified, use its index to find the corresponding `docid`
   and `reason` in the `Parameter Value`.
   - Locate the specific data in `Previous Result` based on the `docid` and ensure
   the data matches the reasons and conditions for use.
   - The results from `Previous Result` (API1) will be applied to the parameters in
   the current API call (API2) following the explanations in the `reason`.
3. **Populate the Dictionary**:
   - Create a dictionary where each parameter name (from the `Parameter Dictionary`) is
   the key, and the extracted value from `Previous Result` is the corresponding value.
   - Populate only those parameters that are explicitly mentioned in the
   `Parameter Value`.
   Exclude all others.
   - **DO NOT use any default values or other values from the `API Document` to populate
   parameters.**
4. **Validate and Output**:
   - Confirm that all parameters listed in the `Parameter Value` are properly populated
   without using default or unrelated values from the `API Document`.
   - Return a dictionary where each parameter name is the key and the parameter value is
   the value of the dictionary.
   - If no parameters can be properly populated using the provided data and reasons,
   return an empty dictionary.
ONLY return the parameter dictionary as your output. DO NOT include any other words.

**Make Call Step 2**

```
You are an API Documentation Assistant responsible for completing function call parameters
based on the API documentation and a partially filled parameter dictionary.
You will be provided with:
1. `API Document`: A dictionary containing information about the API function, including
   its details, required parameters, optional parameters, and their respective default
   values.
2. `Partially Filled Parameters`: A dictionary where some parameters have already been
   populated, but others are still missing.
Your task is to:
1. Review the `API Document` to identify which parameters (required and optional) are
   still missing from the `Partially Filled Parameters` dictionary.
2. Populate the missing parameters based on the following rules:
   - Fill in missing parameters with appropriate values that align with the parameter
     descriptions in the `API Document`. Use your judgment to select realistic and
     suitable values.
   - Ensure all required parameters are included with appropriate values.
   - Optional parameters can remain unfilled if no suitable value can be determined.
3. Ensure that all parameter values match the correct data types specified in the
   `API Document`.
Output Format:
- Return a dictionary where each parameter name is the key, and the parameter value is
  the value of the dictionary.
- The dictionary must include all required parameters (filled with appropriate values)
  and may include optional parameters (if filled).
- Do not include any other words or explanations in the output.
ONLY return the completed parameter dictionary as your output.
```

Figure 17: **API call sequence generation prompt.**

## Sub-instruction Prompt

```
You are an instruction generation assistant for generating lanuage instruction
that enables execution of given API call.
You will be provided a dictionary containing the following keys:
1. 'API Document': A structured description of the API, including its purpose,
   required and optional parameters, and any relevant context about its functionality.
2. 'API call': A dictionary of specific parameter values intended for execution of
   the API call. You must generate language instruction that enables execution of
   this call.
3. 'Previous API Response': The output or result from preceding API calls.
   Some values in 'API call' references values in this result. If this is empty,
   it should not be referenced.
### Your task is to follow these steps:
1. ** Classify Parameters in 'API call':
     - For each key in 'API call', check if its value can be directly derived from
       the 'Previous API Response'.
     - Classify keys into two groups:
         a. Derived Parameters: Parameters whose values are obtained from
         the 'Previous API Response'.
         b. Fixed Parameters: Parameters with values that are not contained
         in 'Previous API Response'.
2. ** Generate Language Instruction:
     - Generate a clear and concise language instruction that enables the execution of
       the 'API call'.
     - Use the 'API Document' to understand the intent of the 'API call' and ensure that
       the generated instruction aligns with its goal. The instruction must be
       goal-oriented, actionable, and contextually accurate.
     - Incorporate the parameter classification:
         a. Derived Parameters: For parameters classified as derived, include in
             the instruction a detailed explanation of how their values are obtained from
             the 'Previous API Response.' Clearly reference the specific part or context of
             the 'Previous API Response' used to derive these values.
             Do not include the derived value itself in the instruction. Instead,
             describe the reasoning behind its selection, such as it being the first item,
             the most recent value, the largest attribute, or another logical criterion.
             The reasoning must be explicit and actionable.
         b. Fixed Parameters: For parameters classified as fixed, include their
             specific values directly in the instruction. Ensure these values are
             explicitly stated to avoid ambiguity.
     - The instruction should naturally integrate both types of parameters and describe
       the action to be performed in a clear and executable manner.
Output Format:
- Return a dictionary with the following keys:
     - "thought": Provide a brief but clear explanation of your reasoning process,
       including how parameters were classified, how derived values were selected,
       and how they were incorporated into the instruction.
     - "instruction": Generate a concise, goal-oriented sentence that describes
       the action required to execute the given API call. Ensure the instruction
       integrates both derived and fixed parameters appropriately, specifying
       derived parameter contexts, the reasoning for their selection, and explicitly
       stating fixed parameter values.
DO NOT use the vague terms such as "use the obtained value" or "from specific values."
DO NOT include parameter names or technical jargon from the API.
Translate these into natural language descriptions of their role or value.
ONLY return the output dictionary. DO NOT include any other words.
```

Figure 18: **Sub-instruction generation prompt.**

1620
1621
1622
1623
1624
1625
1626
1627
1628
1629
1630
1631
1632
1633
1634
1635
1636
1637
1638
1639
1640
1641
1642
1643
1644
1645
1646
1647
1648
1649
1650
1651
1652
1653
1654
1655
1656
1657
1658
1659
1660
1661
1662
1663
1664
1665
1666
1667
1668
1669
1670
1671
1672
1673

### Query Prompt

```
You are an API Documentation Assistant. Your role is to interpret a list of
subinstructions|understanding that each subinstruction is planned, executed,
and possibly leads to creating or adapting subsequent subinstructions based on
prior outcomes|and convert them into a single, high-level user query that reflects
their collective intent without revealing any internal steps or technical API jargon.
### Provided Information:
- Subinstructions: A sequence of iterative steps working toward a single overarching
  objective. They are planned and executed in order, and each result can influence the
  creation or modification of the next subinstruction.
### Your Task:
1. Infer the broader purpose by analyzing how these subinstructions connect logically
   and build upon each other's results.
2. Synthesize them into one natural, user-friendly query that preserves crucial details
   and dependencies but does not mention the subinstructions themselves.
3. Represent information at a high level wherever possible, but retain all specific
   details
   (e.g., IDs, names, dates) from the **first subinstruction** exactly as they are.
4. For subinstructions after the first one, prioritize connecting them through context
   (e.g., "first video," "latest episode") rather than using specific identifiers
   unless absolutely necessary.
5. Ensure that every subinstruction meaningfully contributes to the final query,
   preventing any extraneous or unaligned steps.
6. Avoid any technical language or references to specific APIs in the final query.
### Guidelines:
- Include all essential identifiers or conditions (e.g., names, dates, relevant context)
  from the subinstructions. Do not omit or generalize key details from the
**first subinstruction**.
- For subsequent subinstructions, derive necessary information from the results of prior
  steps whenever possible. Use contextual references to maintain continuity without
  repeating specific identifiers unless required.
- Reflect the necessary sequence or dependency implied by the iterative nature of
  subinstructions without explicitly describing each step.
- The final query must reflect the intent of **all subinstructions** to ensure no step
  becomes irrelevant or disconnected.
### Output Format:
- Return only a dictionary with two keys:
- "thought": A short explanation of how you derived the final query from the
  subinstructions.
- "query": The single-sentence user query representing the overall goal.
DO NOT include API names or technical jargon from the API.
ONLY return the dictionary as your output. DO NOT include any other words.
```

Figure 19: **User query generation prompt.**

---

**Final Response Prompt**

```
You are an answer generation assistant tasked with providing natural language responses
to user queries by analyzing API call results.
You will be provided a dictionary containing the following keys:
1. 'User Query': A natural language question or request from the user.
2. 'API Call Result': A list of dictionaries, each representing a step or subinstruction
   carried out to fulfill the user query. Each dictionary contains:
- 'subinstruction': A brief description of the step taken.
- 'api response': The actual data or result obtained from executing the subinstruction.
### Your task is to follow these steps:
1. ** Analyze API Call Result: **
- Examine each dictionary in the 'API Call Result' list.
- Understand the purpose of each 'subinstruction' and the corresponding 'api response.'
- Identify how each 'api response' contributes to answering the 'User Query.'
- If necessary, combine results from multiple subinstructions to generate a comprehensive
  answer.
2. ** Generate Final Answer: **
- Construct a coherent and natural response to the 'User Query' based on the collected
  information from 'API Call Result.'
- Use clear and concise language, phrasing the answer in a way that feels conversational
  and human-like.
- Ensure the final response directly addresses the user's request without unnecessary
  detail.
- Summarize or filter the results if needed, prioritizing the most relevant information.
- If any subinstruction does not yield meaningful data, exclude it from the final answer
  and focus on the most relevant results.
Output Format:
- Return a dictionary with the following keys:
- "thought": Provide a concise summary of how the API Call Result was analyzed,
  how relevant subinstructions were chosen, and how they were combined to address
  the User Query.
- "final_answer": A natural language response to the User Query, synthesized from the
  API Call Result.
This should sound as if an agent is directly responding to the user.
DO NOT include API jargon or technical terms in the final answer.
Only present information in user-friendly, natural language.
Focus on delivering the information as if you are the final point of communication with
the user.
ONLY return the output dictionary. DO NOT include any other words.
```

Figure 20: **Final response generation prompt.**

---

**Success Rate Prompt**

```
Given a user query, a sequence of tool execution details (including successes and
failures),
and the final answer, determine whether the answer sufficiently and correctly solves
the original query, strictly based on the tool execution results.
Evaluation Rules:
1. The final answer must be based on the tool execution results.
- If the answer is generated independently without using the tool results, return
  "Unsolved".
2. The final answer must address and resolve **all parts** of the user query.
  Partial answers are not accepted.
- If the answer does not fully respond or give valid answer to every part of the query,
  return "Unsolved".
3. Only if the answer is fully based on tool results **and** correctly answers all
  aspects of the query, return "Solved".
  No "Unsure" status is allowed.
Output format:
{
"content": "<Step-by-step reasoning and explanation>",
"answer_status": "Solved" | "Unsolved"
}
```

Figure 21: **Prompt used for success rate metric.**

