# OpenReview forum: "GOAT: A Training Framework for Goal-Oriented Agent with Tools"
_ICLR.cc/2026/Conference — ICLR 2026 Conference Withdrawn Submission_

### Official Review · Reviewer_Gkbm · 2025-10-27

**Soundness:** 1
**Presentation:** 2
**Contribution:** 1
**Rating:** 2
**Confidence:** 5

**Summary:**

This paper identifies a challenge in developing tool-using LLM agents: open-source models struggle with "goal-oriented" queries that require complex planning and multiple, interdependent API calls. The authors attribute this to a lack of suitable training data.

To solve this, they propose GOAT (Goal-Oriented Agent with Tools), a training framework that automatically generates synthetic, goal-oriented training data from API documentation without human annotation. The core of GOAT is a three-stage pipeline:
1.  API Dependency Graph Construction: It parses API documents and builds an initial graph of all possible input-output connections.
2.  Graph Refinement: This graph is progressively filtered for feasibility using three steps: embedding similarity, LLM-based semantic checking, and finally, validation via actual API call execution.
3.  Data Generation: The framework samples connected subgraphs (valid API sequences) and uses a "call-first" strategy. It first executes the API sequence, then generates corresponding sub-queries, and finally generates a high-level user query and a final response based on the execution trace.

The authors demonstrate that fine-tuning open-source models on this GOAT-generated data improves performance on several goal-oriented benchmarks, including RestBench, API-Bank, and a new benchmark introduced in this paper, GOATBench.

**Strengths:**

1.  Sensible "Call-First" Design: The "call-first" generation strategy (Section 3.1.3) is a sound design choice. Generating a natural language query from a concrete, executable API path is a much more constrained and reliable task for an LLM than the reverse (generating correct API calls from a vague query), likely leading to higher-fidelity training data.

2.  Contribution of a New Benchmark: The introduction of GOATBench, a human-verified benchmark for goal-oriented tasks, is a useful contribution to the community, which currently lacks diverse and challenging evaluation sets for this specific problem.

**Weaknesses:**

1.  Outdated and Weak Baselines in Key Comparisons: The experimental setup is undermined by the use of outdated models and weak baselines in key comparative tables.
    * In Table 2, the comparisons are against text-davinci-003 (a 2022 model) and open-source models like Llama2-13B and Vicuna-13B. These are not competitive backbones for a paper dated late 2025.
    * Similarly, Table 3's baselines (Alpaca-7B, Llama-7B) are obsolete.
    * The paper lacks a crucial comparison: training a model using data generated by another framework (like ToolFlow) and evaluating it on GOATBench, which would be a true test of GOAT's data generation superiority.

2.  Limited Methodological Novelty: The paper's core pipeline (build a dependency graph, filter it, sample paths, and generate data) is not new. The related work section and Table 1 already cite several highly similar, contemporaneous works (e.g., ToolFlow, Magnet, ToolDial) that also use graph-based methods to synthesize agent data. The 3-stage filtering pipeline (SBERT -> LLM check -> Execution) is a pragmatic engineering workflow, not a fundamental research contribution. The "call-first" strategy is also a known approach. The paper's primary contribution seems to be the application of this existing template to single-turn "goal-oriented" queries, which feels incremental.

3. The author didn't open-source GOATBench.

**Questions:**

Please solve the weakness provided.

---

> ### Author Response · Authors · 2025-11-21
>
> We appreciate the reviewer’s thorough assessment and insightful suggestions. We respond to each issue individually below.
>
> &nbsp;&nbsp;
> &nbsp;&nbsp;&nbsp;
>
> > 1-1. Outdated and Weak Baselines in Key Comparisons: The experimental setup is undermined by the use of outdated models and weak baselines in key comparative tables.
> In Table 2, the comparisons are against text-davinci-003 (a 2022 model) and open-source models like Llama2-13B and Vicuna-13B. These are not competitive backbones for a paper dated late 2025.
> Similarly, Table 3's baselines (Alpaca-7B, Llama-7B) are obsolete.
>
> We clarify that the older baselines were not chosen by preference, but because RestBench [1] and API-Bank [2] **do not release reproducible code**, making it impossible to re-run their baselines on newer models. Although we re-implemented their evaluation pipelines faithfully following the original papers, the reproduced systems produced near-zero scores, falling far short of the performance reported in the original publications. Therefore, we followed the standard practice of **directly reporting the results published in those benchmark papers**, which necessarily rely on older models.
>
> For a fair comparison, in Table 2 and 3 in Section 4.3, we applied our method to the same backbone models used in the original benchmarks and measured the improvements introduced by GOAT. To further address your concern, we additionally conducted experiments with **modern backbones (Llama-3-8B-Instruct and Qwen-2.5-7B-Instruct)** and observed similarly strong gains from GOAT, as shown below.
>
> | Backbone     | Training Method | TMDB Success % | Spotify Success % |
> |--------------|------------------|----------------|--------------------|
> | Llama-3-8B   | Zero-shot        | 1              | 10.5               |
> | Llama-3-8B   | GOAT finetuned   | **14**         | **19.3**           |
> | Qwen-2.5-7B  | Zero-shot        | 0              | 8.8                |
> | Qwen-2.5-7B  | GOAT finetuned   | **6**          | **29.8**           |
>
> *Experiment in RestBench. All experiments in this table use the Baseline prompting method.
>
> | Backbone     | Training Method   | Success % |
> |--------------|--------------------|-----------|
> | Llama-3-8B   | Zero-shot          | 34        |
> | Llama-3-8B   | GOAT finetuned     | **64**    |
> | Qwen-2.5-7B  | Zero-shot          | 28        |
> | Qwen-2.5-7B  | GOAT finetuned     | **66**    |
>
> *Experiment in API-Bank (Plan+Retrieve Call). All experiments in this table use the Baseline prompting method.
>
> [1] Song, Yifan, et al. "Restgpt: Connecting large language models with real-world restful apis." arXiv preprint arXiv:2306.06624 (2023).
>
> [2] Li, Minghao, et al. "API-Bank: A Comprehensive Benchmark for Tool-Augmented LLMs." Proceedings of the 2023 Conference on Empirical Methods in Natural Language Processing. 2023.

---

> > ### Author Response · Authors · 2025-11-21
> >
> > > 1-2. The paper lacks a crucial comparison: training a model using data generated by another framework (like ToolFlow) and evaluating it on GOATBench, which would be a true test of GOAT's data generation superiority.
> >
> > We appreciate the reviewer’s suggestion. **Table 4 in Section 4.3**, already shows that models trained on multi-step instruction data (ToolLLM [3]) transfer poorly to goal-oriented tasks, while GOAT-trained models achieve substantial gains.
> >
> > To further examine this point, we tested data generation from **ToolFlow [4]** and **Magnet [5]** in API-Bank. Both methods were originally designed for multi-turn, multi-step instruction generation, so we merged all steps into a single instruction and paraphrased it into a one-turn goal-oriented query. ToolFlow relies only on similarity-based filtering and Magnet relies only on LLM-based filtering, yielding **low-precision dependency graphs** with many invalid edges. Moreover, ToolFlow **fabricates tool outputs** instead of performing real API calls, so the produced outputs often cannot be used by subsequent APIs, breaking the dependency chain. Magnet’s instruction-first generation is not grounded in real API executions, so the generated instructions often omit essential argument values or misinterpret API’s functionality, resulting in **hallucinated instructions and failed API invocations.**
> >
> > As a result, models trained on ToolFlow- or Magnet-generated data showed only marginal improvements over zero-shot performance, remaining far below the gains achieved by GOAT. Notably, we generated ToolFlow and Magnet data using GPT-4.1, whereas GOAT’s synthetic data was produced using the smaller Llama-3-70B. Despite relying on a much stronger LLM, ToolFlow and Magnet still failed to produce usable goal-oriented data.
> >
> > *ToolDial [6] focuses on conversational dynamics such as clarifications and system-level interaction states, making it unsuitable for conversion into a one-shot goal-oriented setup; thus we excluded it from evaluation.
> >
> > | Data generation Method | Success % |
> > |------------------------|-----------|
> > | Zero-shot              | 28        |
> > | Toolflow               | 30        |
> > | Magnet                 | 32        |
> > | GOAT                   | **66**    |
> >
> > *Experiment in API-Bank (Plan+Retrieve Call). All experiments in this table use the Baseline prompting method with a Qwen-2.5-7B backbone.
> >
> > [3] Qin, Yujia, et al. "ToolLLM: Facilitating Large Language Models to Master 16000+ Real-world APIs." ICLR. 2024.
> >
> > [4] Wang, Zezhong, et al. "Toolflow: Boosting llm tool-calling through natural and coherent dialogue synthesis." Proceedings of the 2025 Conference of the Nations of the Americas Chapter of the Association for Computational Linguistics: Human Language Technologies (Volume 1: Long Papers). 2025.
> >
> > [5] Yin, Fan, et al. "Magnet: Multi-turn tool-use data synthesis and distillation via graph translation." Proceedings of the 63rd Annual Meeting of the Association for Computational Linguistics (Volume 1: Long Papers). 2025.
> >
> > [6] Shim, Jeonghoon, et al. "ToolDial: Multi-turn Dialogue Generation Method for Tool-Augmented Language Models." The Thirteenth International Conference on Learning Representations.

---

> > > ### Author Response · Authors · 2025-11-21
> > >
> > > > 2. Limited Methodological Novelty: The paper's core pipeline (build a dependency graph, filter it, sample paths, and generate data) is not new. The related work section and Table 1 already cite several highly similar, contemporaneous works (e.g., ToolFlow, Magnet, ToolDial) that also use graph-based methods to synthesize agent data. The 3-stage filtering pipeline (SBERT -> LLM check -> Execution) is a pragmatic engineering workflow, not a fundamental research contribution. The "call-first" strategy is also a known approach. The paper's primary contribution seems to be the application of this existing template to single-turn "goal-oriented" queries, which feels incremental.
> > >
> > > Thank you for the thoughtful assessment. We would like to clarify three aspects in which GOAT makes distinct and novel contributions.
> > >
> > > **1. First framework that automatically generates training data for goal-oriented queries.**
> > >
> > > GOAT is, to our knowledge, the first system to automatically construct goal-oriented training data where implicit user query requires: (i) inferring which APIs to call and in what order, and (ii) filling each API’s parameters according to dependency propagation— without any explicit step-wise descriptions.
> > > In contrast, existing systems such as ToolFlow, Magnet, and ToolDial operate in multi-step and multi-turn settings where: (i) the API sequence is explicitly spelled out in the instruction, and (ii) API calls are often independent and do not require dependency reasoning.
> > >
> > > As our experiment above shows, these methods fail to generate usable data for goal-oriented evaluation (ToolFlow and Magnet yield only marginal gains over zero-shot and remain far below GOAT),  illustrating that this problem is **far from a trivial extension of prior templates**.
> > >
> > > **2. Execution-based dependency validation for high-precision graphs.**
> > >
> > > GOAT introduces a three-stage filtering pipeline with an execution-based validation that raises edge precision to 90% (Table 7 in Appendix A.2). Prior frameworks such as ToolFlow, Magnet, and ToolDial do not execute real APIs during filtering, leaving many invalid edges in their dependency graphs and consequently producing large numbers of unusable or incorrect trajectories—as also reflected in our experiment above.
> > >
> > > For goal-oriented tool use, dependency fidelity is critical: a single incorrect edge invalidates the entire multi-step chain. Therefore, rigorous filtering grounded in real API execution is not an engineering detail but a task-specific necessity for generating executable trajectories at scale.
> > >
> > > **3. A call-first pipeline that grounds trajectories before generating instructions.**
> > >
> > > Unlike prior work that follows an instruction-first paradigm—where they first generates an instruction from API documentation and then annotates an API call path through LLM inference—GOAT uses a call-first formulation, iteratively executing the APIs first and then generating the instruction afterward based on the grounded execution results. As the reviewer also acknowledged in **Strength 1**, this leads to **significantly higher-fidelity training data**. To our knowledge, no existing work generates natural-language instructions after constructing a fully grounded trajectory through iterative real API execution. If the reviewer is aware of any such prior system, we would very much appreciate the reference and will carefully examine it and provide a sincere, detailed response.
> > >
> > > &nbsp;&nbsp;
> > > &nbsp;&nbsp;&nbsp;
> > >
> > > > 3. The author didn't open-source GOATBench.
> > >
> > > Thank you for pointing this out. We fully agree that releasing GOATBench is important for reproducibility. In the camera-ready version, we will publicly release the **entire GOAT suite**, including: (1) GOATBench dataset, (2) GOAT data-generation pipeline, (3) training and evaluation code, and (4) all fine-tuned model checkpoints. The full open-source repository is already being prepared and will be available upon publication.

---

> ### Comment · Reviewer_Gkbm · 2025-11-25
>
> I would like to thank the authors for their detailed response and the effort put into the additional experiments with modern backbones.
>
> However, the rebuttal has not fully alleviated my concerns regarding the methodological novelty. While I acknowledge the distinction the authors drew between "goal-oriented" and "multi-step" tasks, the proposed GOAT framework appears to primarily extend existing synthetic data generation paradigms by incorporating an API-documentation-based setup. As the target is multi-tool use, one would expect the contribution to be weighed against the broader state-of-the-art in agentic frameworks. From this perspective, the method feels more like an incremental adaptation of data synthesis pipelines rather than an exciting pushup.
>
> Regarding reproducibility, I appreciate the authors' commitment to open-sourcing the GOAT suite. Given the complexity of the pipeline, I strongly encourage the authors to release the full codebase and benchmark data upon submission to allow the community to verify the reported performance.
>
> After carefully considering the rebuttal and the paper's contributions, I will maintain my original rating.

---

### Official Review · Reviewer_Bw8H · 2025-10-29

**Soundness:** 3
**Presentation:** 3
**Contribution:** 2
**Rating:** 4
**Confidence:** 4

**Summary:**

This paper proposes GOAT (Goal-Oriented Agent with Tools), a human annotation-free training framework designed to enhance the capability of open-source large language models (LLMs) on goal-oriented tool-use tasks. In such tasks, the model must autonomously decompose a high-level user goal, plan a multi-step API call chain, reason about inter-API parameter dependencies, and generate a final natural-language response.

**Strengths:**

- The work targets the realistic and highly challenging setting of goal-oriented tool use, clearly identifying the key bottleneck for open-source models: the lack of training data for multi-step, interdependent API calls. This addresses a clear practical need.
- GOAT-trained agents consistently outperform zero-shot and existing fine-tuned baselines across multiple benchmarks, including RestBench, API-Bank, and the newly introduced GOATBench.
- The paper provides detailed prompt templates, hyperparameters, and data examples in the appendix, significantly supporting reproducibility.

**Weaknesses:**

- The method assumes that API documentation contains clear natural language descriptions of parameters and outputs. However, in practice, many real-world APIs—especially internal or niche services—have poor-quality, incomplete, or unstructured documentation. Under such conditions, the API dependency graph construction would likely fail. The authors do not evaluate robustness under low-quality or unstructured documentation.
- The framework supports only DAG-structured API sequences with up to 4 nodes, and cannot handle cyclic dependencies, conditional branching (e.g., if-else), or error recovery. Moreover, the inference process follows a static plan-then-execute paradigm and does not integrate iterative reasoning (e.g., ReAct-style reflection) or mechanisms for handling API call failures.
- The three-stage filtering pipeline—especially the API call execution validation step—involves numerous real API invocations and LLM inferences, which could incur significant time and monetary costs when scaling to large API sets.
- The definition of “goal-oriented” is ambiguous, and the boundary with multi-step instruction tasks is unclear. For example, generated queries like *“Can you help me find a hotel near the Flixbus stations in Paris?”* already imply a task structure (first locate stations, then find hotels), resembling implicit multi-step instructions. The distinction from prior work like API-Bank may thus reflect only a difference in phrasing granularity, rather than a fundamental innovation.

**Questions:**

- Is the superiority of the “call-first” strategy empirically validated? The authors claim this approach avoids the self-reinforcing bias of instruction-first methods, but no ablation or comparative experiment is provided (e.g., training on instruction-first vs. call-first data using the same APIs and model).
- GOATBench’s “unseen” split only involves unseen tool compositions within the same domains (finance, food, entertainment, travel). Can the framework generalize to entirely new domains (e.g., healthcare or legal APIs) that were absent during training?
- Are the retriever (SBERT) and LLM (LoRA) jointly fine-tuned end-to-end, or trained separately? If trained in separate stages, could error propagation (e.g., incorrect API retrieval leading to failed planning) undermine overall performance?

---

> ### Author Response · Authors · 2025-11-21
>
> Thank you for the careful evaluation and constructive comments. We address each point in detail below.
>
> > 1. The method assumes that API documentation contains clear natural language descriptions of parameters and outputs. However, in practice, many real-world APIs—especially internal or niche services—have poor-quality, incomplete, or unstructured documentation. Under such conditions, the API dependency graph construction would likely fail. The authors do not evaluate robustness under low-quality or unstructured documentation.
>
> We thank the reviewer for the valuable comment. We clarify that GOAT does not rely on clean or formally structured API documentation. As described in Section 3.1.1 (L198), we employ an LLM to interpret documentation and extract parameter and output descriptions. This design enables GOAT to function even when the original documentation is noisy, informal, unstructured, or inconsistent. Moreover, even when documentation is unavailable, existing techniques that translate source code into textual descriptions [1] can be integrated without modification to our method.
>
> Importantly, if the documentation is excessively noisy to the extent that it fails to convey any meaningful semantics, such a failure would fundamentally affect all components of the agent—including retrieval augmentation—since the absence of usable documentation prevents identifying which API to invoke, rendering the tool-learning task itself infeasible regardless of GOAT.
>
> [1] Feng et al., “CodeBERT: A Pre‑Trained Model for Programming and Natural Languages”, EMNLP,  2020.
>
> &nbsp;&nbsp;
> &nbsp;&nbsp;&nbsp;
>
> > 2. The framework supports only DAG-structured API sequences with up to 4 nodes, and cannot handle cyclic dependencies, conditional branching (e.g., if-else), or error recovery. Moreover, the inference process follows a static plan-then-execute paradigm and does not integrate iterative reasoning (e.g., ReAct-style reflection) or mechanisms for handling API call failures.
>
> **1. On DAG-only constraint.**
> We clarify that our framework is not inherently restricted to DAG-structured API sequences. The current implementation focuses on acyclic tool-use workflows simply because such patterns were predominant in our observed data. Cyclic dependencies can be incorporated by adjusting the subgraph sampling strategy without requiring architectural changes. Exploring realistic cyclic tool-use scenarios and extending GOAT to such cases would indeed be an interesting direction for future work.
>
> **2. On inference-time iterative reasoning.**
>
> GOAT is not limited to a static plan-then-execute paradigm. It generates complete API invocation trajectories that can be directly utilized by different agent architectures, including those employing iterative reasoning. As shown in Tables 4 and 8, ReAct-style models trained with GOAT-generated data outperform their corresponding baselines, demonstrating that the framework readily supports iterative reasoning settings.
>
> **3. On conditional branching and error handling.**
>
> We thank the reviewer for raising this constructive point. We agree that modeling conditional branching, error recovery, and other dynamic behaviors arising during execution is an important next step. These behaviors extend beyond deterministic supervision and require adaptive control policies. We view extending GOAT to handle such runtime dynamics as a valuable and promising direction for future work.
>
> &nbsp;&nbsp;
> &nbsp;&nbsp;&nbsp;
>
> > 3. The three-stage filtering pipeline—especially the API call execution validation step—involves numerous real API invocations and LLM inferences, which could incur significant time and monetary costs when scaling to large API sets.
>
> We thank the reviewer for the concern. While the execution-validation step introduces some additional cost during graph construction, it reduces the overall cost of the full data-generation pipeline. After the dependency graph is built, the pipeline repeatedly samples subgraphs and executes API calls to generate queries. If the graph contains an invalid edge, the same inexecutable API call will be triggered across many sampled subgraphs, causing all of them to fail. This leads to repeated API invocations and compounding cost during large-scale generation.
>
> GOAT avoids this issue by performing execution-validation up front, filtering out invalid API connections before subgraph sampling. This drastically reduces the number of invalid subgraphs encountered later, preventing the repeated execution of inexecutable API calls and avoiding large-scale wasted inference.
>
> Thus, **the proposed filtering pipeline is cost-saving by design**: it introduces a small, controlled one-time overhead to eliminate far larger multipliers in API-call cost when generating thousands of subgraphs.

---

> > ### Author Response · Authors · 2025-11-21
> >
> > > 4-1. The definition of “goal-oriented” is ambiguous, and the boundary with multi-step instruction tasks is unclear. For example, generated queries like “Can you help me find a hotel near the Flixbus stations in Paris?” already imply a task structure (first locate stations, then find hotels), resembling implicit multi-step instructions.
> >
> > Thank you for raising this question. We clarify that GOAT’s goal-oriented setting is fundamentally different from typical multi-step instruction tasks, and this distinction goes beyond phrasing.
> >
> > **Multi-step instructions** explicitly specify both the operations and their order. For example, a TaskBench [2] instruction states: *“I want to convert a landscape video (video.mp4) into an anime style, then dub the video based on a story (script.txt). Finally, I want to post the newly created video on my TikTok.”* Here, the plan is fully spelled out; the model should simply map each step to the corresponding API.
> >
> > **Goal-oriented queries**, in contrast, provide only a high-level objective and require the model to **infer the full dependency chain**. For example, the GOATBench query, *“Get the comments and related videos for the trending video in Spain.”*, demands deducing: (1) *retrieve the trending video list*, (2) *query with video name to get the video ID* (3)  *fetch the video comments* and (4) *related videos using that ID*—none of which are explicitly stated. Moreover, the decomposition itself varies with the API set: if IDs appear in step (1), step (2) is unnecessary; if comments are embedded in the video-detail API, step (3) changes into a video-detail retrieval call. Thus, **solving a goal-oriented query is inherently non-trivial**: the model must infer not only the correct sequence but also how to adapt the plan to the specific API schema and dependency structure.
> >
> > Differences appear empirically as well. Models trained on multi-step instruction data generated via ToolLLM [3] achieve only about half the performance of GOAT-trained models on GOATBench (Table 4). We additionally evaluated multi-step data generation methods (ToolFlow [4] and Magnet [5]) on API-Bank[6]; both yielded only marginal gains over non-fintuned result, far below GOAT. These results show that **multi-step supervision is insufficient for teaching goal-oriented inference**, confirming that GOAT addresses a substantively different and more challenging setting.
> >
> > | Data generation Method | Success % |
> > |------------------------|-----------|
> > | Zero-shot              | 28        |
> > | Toolflow               | 30        |
> > | Magnet                 | 32        |
> > | GOAT                   | **66**    |
> >
> > *Experiment in API-Bank (Plan+Retrieve Call). All experiments in this table use the Baseline prompting method with a Qwen-2.5-7B backbone.
> >
> > [2] Shen, Yongliang, et al. "Taskbench: Benchmarking large language models for task automation." Advances in Neural Information Processing Systems 37 (2024): 4540-4574.
> >
> > [3] Qin, Yujia, et al. "ToolLLM: Facilitating Large Language Models to Master 16000+ Real-world APIs." ICLR. 2024.
> >
> > [4] Wang, Zezhong, et al. "Toolflow: Boosting llm tool-calling through natural and coherent dialogue synthesis." Proceedings of the 2025 Conference of the Nations of the Americas Chapter of the Association for Computational Linguistics: Human Language Technologies (Volume 1: Long Papers). 2025.
> >
> > [5] Yin, Fan, et al. "Magnet: Multi-turn tool-use data synthesis and distillation via graph translation." Proceedings of the 63rd Annual Meeting of the Association for Computational Linguistics (Volume 1: Long Papers). 2025.
> >
> > [6] Li, Minghao, et al. "API-Bank: A Comprehensive Benchmark for Tool-Augmented LLMs." Proceedings of the 2023 Conference on Empirical Methods in Natural Language Processing. 2023.
> >
> > &nbsp;&nbsp;
> > &nbsp;&nbsp;&nbsp;
> >
> > > 4-2. The distinction from prior work like API-Bank may thus reflect only a difference in phrasing granularity, rather than a fundamental innovation.
> >
> > While API-Bank is indeed goal-oriented, this applies only to its human-annotated test set. Its training data, however, are constructed by randomly sampling APIs and concatenating them into a single query, without verifying functional dependencies among them. As a result, the training set does not encode compositional structure—one API’s output is not guaranteed to serve as the input to another—and thus fails to represent executable, goal-oriented reasoning.
> >
> > In contrast, GOAT explicitly models dependency-driven API trajectories, where each call is grounded in the outputs of preceding ones. This enables learning of compositional reasoning and executable multi-step planning, which are essential for realistic tool use. To our knowledge, no prior framework automatically generates goal-conditioned queries with their corresponding executable dependency chains. Thus, GOAT introduces a substantive conceptual advancement, not merely a difference in phrasing granularity.

---

> > > ### Author Response · Authors · 2025-11-21
> > >
> > > > 5. Is the superiority of the “call-first” strategy empirically validated? The authors claim this approach avoids the self-reinforcing bias of instruction-first methods, but no ablation or comparative experiment is provided (e.g., training on instruction-first vs. call-first data using the same APIs and model).
> > >
> > > We thank the reviewer for raising this important point. To directly validate the effectiveness of the call-first strategy, we compared it with an instruction-first variant within the same pipeline, using identical subgraphs in RestBench[7]. While both approaches started with the same number of TMDB and Spotify subgraphs, the instruction-first pipeline mostly produced non-executable instances, yielding only 537 and 69 valid samples, respectively. In contrast, the call-first strategy successfully generated 8,570 (TMDB) and 924 (Spotify) valid instances by directly constructing executable API calls. As shown in the table below, models trained on these valid samples achieved significantly higher performance with the call-first approach, empirically confirming its advantage.
> > >
> > > | Data generation Method | TMDB Success % | Spotify Success % |
> > > |------------------------|----------------|--------------------|
> > > | Instruction-first      | 5              | 17.5               |
> > > | Call-first             | **7**          | **28.1**           |
> > >
> > >
> > > *All experiments in this table use the Baseline prompting method with a Llama-2-13B backbone.
> > >
> > > [7] Song, Yifan, et al. "Restgpt: Connecting large language models with real-world restful apis." arXiv preprint arXiv:2306.06624 (2023).
> > >
> > > &nbsp;&nbsp;
> > > &nbsp;&nbsp;&nbsp;
> > >
> > > > 6. GOATBench’s “unseen” split only involves unseen tool compositions within the same domains (finance, food, entertainment, travel). Can the framework generalize to entirely new domains (e.g., healthcare or legal APIs) that were absent during training?
> > >
> > > We thank the reviewer for the insightful question. To assess cross-domain generalization, we trained the model on GOATBench and evaluated it on RestBench, which contains entirely different APIs and domains. As shown in the table below, the model achieves non-trivial improvements in success rate on both TMDB and Spotify compared to the non-finetuned baseline, demonstrating that GOAT-trained models generalize effectively beyond the API domains observed during training.
> > >
> > > | Training Method          | TMDB Success % | Spotify Success % |
> > > |--------------------------|----------------|--------------------|
> > > | Zero-shot                | 1              | 10.5               |
> > > | GOAT – GOATBench         | 10             | 14.0               |
> > > | GOAT – RestBench         | 12             | 21.1               |
> > >
> > >
> > > *All experiments in this table use the Baseline prompting method with a Llama-3-8B backbone.
> > >
> > > &nbsp;&nbsp;
> > > &nbsp;&nbsp;&nbsp;
> > >
> > > > 7. Are the retriever (SBERT) and LLM (LoRA) jointly fine-tuned end-to-end, or trained separately? If trained in separate stages, could error propagation (e.g., incorrect API retrieval leading to failed planning) undermine overall performance?
> > >
> > > We appreciate the reviewer’s thoughtful question. GOAT provides ground-truth trajectories for all modules of an agent (retrieval targets, plans, and API calls), allowing each component to be supervised directly. Accordingly, we train the retriever and the LLM separately, each using its own ground-truth labels, consistent with prior tool-use frameworks [2]. While we agree that joint end-to-end optimization could potentially mitigate error propagation, performing such optimization would require a non-trivial yet orthogonal extension to our training sample generation pipeline. We therefore regard this as an interesting direction for future work.
> > >
> > > [2] Shen, Yongliang, et al. "Taskbench: Benchmarking large language models for task automation." Advances in Neural Information Processing Systems 37 (2024): 4540-4574.

---

### Official Review · Reviewer_4MzE · 2025-10-31

**Soundness:** 2
**Presentation:** 3
**Contribution:** 3
**Rating:** 4
**Confidence:** 4

**Summary:**

This paper addresses the limitations of large language model (LLM) agents in handling goal-oriented tool-use tasks—specifically, the lack of annotated training data and poor performance of open-source models. The authors propose GOAT, a human-annotation-free training framework that automatically constructs synthetic datasets for goal-oriented API execution tasks from API documents. GOAT first builds a refined API dependency graph via a three-stage filtering pipeline (embedding similarity, LLM reasoning, and real API execution), then samples connected subgraphs to generate task samples (including user queries, interdependent API sequences, and final responses). Additionally, the authors introduce GOATBench, a new benchmark for goal-oriented tasks. Experiments show that GOAT-trained open-source models (e.g., Llama3-8B) achieve state-of-the-art performance on RestBench, API-Bank, and GOATBench, with some metrics even approaching closed-source models like text-davinci-003. The core contributions include: (1) an automatic data generation pipeline for goal-oriented tasks; (2) consistent performance gains across diverse LLMs and benchmarks; (3) the new GOATBench to facilitate future evaluation.

**Strengths:**

**Originality**
This work demonstrates originality in two folds: (1) the authors proposes to generate training data of tool using sequences based on a refined API dependency Graph. The graph construction consists of three stages of filtering which ensures the correct dependencies in the graph. (2) GOAT achieves comparable performance with closed-source models in tool use by finetuning small-scale open-source models with synthetic data.

**Quality & Clarity**
The framework is well-structured, with clear mathematical and procedural descriptions for the main components including API dependency Graph construction, Goal-Oriented API Execution Data Construction. Experiments cover three benchmarks (RestBench, API-Bank, GOATBench) and six LLMs, with both in-domain and out-of-domain evaluations are included. Examples on the synthetic dependency graph and prompt designs details are adequately illustrated in appendices.

**Significance**
This work addresses a critical and practical problem —building cost-effective tool-use agents. The proposed method could enable adaptation of LLM agents to domain-specific APIs (e.g., financial, travel) at low cost without human annotations.

**Weaknesses:**

1. As for the API Dependency Graph construction, unclear about the accuracy of the constructed graph node and edges since the authors proposed to enhance the correctness of the dependency by several strategies. Quantitative analysis on the correctness and noise of dependency graph is expected.

2. The "call-first" strategy fills the API with plausible value at each filed as introduced in Section 3.1.3. However, enumerating all plausible values for every field of each API would constitute a huge mount of data sets. The authors used all the possible data points or they just select/utilize a small portion? There are no detailed discussions on this detail and no experiments to exam how does the selection influence the overall performance.

3. There could be obvious discrepancy between the synthetic training data and the real-world tool use cases. Connected subgraphs are collected to form the synthetic training data as introduced in Section 3.1, but in practice there could be tool calling trajectory on a subgraph with much more nodes (5+ steps v.s. 4 API nodes in this work) and even more some query could involve multiple subgraphs.

4. Lack of comparison with recent closed-source models like GPT-4o or Claude 4 sonnet. Given that GPT-4o has strong zero-shot tool-use capabilities, omitting this comparison weakens the claim that GOAT-trained open-source models are "competitive with closed-source systems."

5. Missing technical details such as the cost of API calls, number of training data points for LLM fine-tuning and number of query-document pairs for SBERT training.

6. Section 3.1.3 seems like discussion on related works, then it would be better to move this paragraph to section 2.

**Questions:**

Several questions and concerns are raised in "Weaknesses" part. I would be willing to change my recommendation according to the authors' response.

---

> ### Author Response · Authors · 2025-11-21
>
> We appreciate the reviewer for the valuable  feedback and suggestions. We address each concern and question point-by-point below.
>
> &nbsp;&nbsp;
> &nbsp;&nbsp;&nbsp;
>
> > 1. As for the API Dependency Graph construction, unclear about the accuracy of the constructed graph node and edges since the authors proposed to enhance the correctness of the dependency by several strategies. Quantitative analysis on the correctness and noise of dependency graph is expected.
>
> To address this concern, we note that **Appendix A.2 Table 7** already provides a quantitative evaluation of edge correctness in the GOATBench training graph. Specifically, we randomly sampled 500 API-to-API edges from all possible connections among the GOATBench APIs and manually annotated each as valid or invalid. After applying each stage of our filtering pipeline, we computed precision and recall by comparing the remaining edges against these manual labels.
>
> | Filtering Step        | Precision | Recall |
> |-----------------------|-----------|--------|
> | Embedding Similarity  | 0.25      | 0.92   |
> | LLM Filtering         | 0.59      | 0.42   |
> | API Execution         | **0.90**      | **0.36**   |
>
> The final execution-grounding stage yields a **high-precision graph (90%)**, which is critical because incorrect edges directly break dependency-driven planning. We intentionally prioritize precision in the final step to ensure that all surviving edges correspond to reliable, executable API connections.
>
> Although recall decreases to 0.36, the resulting graph still retains hundreds of valid edges, which is sufficient for generating a large number of diverse and connected subgraphs, and our experiments support that GOAT-trained models generalize effectively to unseen API connections.
>
> &nbsp;&nbsp;
> &nbsp;&nbsp;&nbsp;
>
> > 2. The "call-first" strategy fills the API with plausible value at each filed as introduced in Section 3.1.3. However, enumerating all plausible values for every field of each API would constitute a huge mount of data sets. The authors used all the possible data points or they just select/utilize a small portion? There are no detailed discussions on this detail and no experiments to exam how does the selection influence the overall performance.
>
> We clarify that we do not enumerate all possible values, nor do we manually define large candidate pools. Instead, **all arguments are selected automatically by the LLM**, guided by: (1) the semantic description in the API documentation, including type hints, examples, and defaults, and (2) the outputs of previous API calls when dependencies exist. This produces plausible, semantically coherent arguments without combinatorial value explosion. Despite using only a single sampled value per argument, we observe strong generalization across three benchmarks (RestBench [1], API-Bank [2], GOATBench).
>
> That said, we agree that augmenting argument values with a predefined value set or sampling a wider distribution could further increase data diversity. To examine this, we conducted a small-scale experiment on the API-Bank (Plan+Call+Retrieve) benchmark, where **we augmented each parameter with three alternative values** during data generation. Even with this lightweight augmentation, we observed improvements in Success Rate:
>
> | Training Method                        | Success % |
> |----------------------------------------|-----------|
> | Zero-shot                                      | 0         |
> | GOAT finetuned                         | 38        |
> | GOAT + value augmentation finetuned    | **40**        |
>
> *All experiments in this table use the Baseline prompting method with a Llama-7B backbone.
>
> These results underscore GOAT’s strong underlying effectiveness, while showing that even a very simple value-level augmentation can yield further gains. This shows that GOAT can improve further when given more training compute for richer augmentation.
>
> &nbsp;&nbsp;
> &nbsp;&nbsp;&nbsp;
>
> [1] Song, Yifan, et al. "Restgpt: Connecting large language models with real-world restful apis." arXiv preprint arXiv:2306.06624 (2023).
>
> [2] Li, Minghao, et al. "API-Bank: A Comprehensive Benchmark for Tool-Augmented LLMs." Proceedings of the 2023 Conference on Empirical Methods in Natural Language Processing. 2023.

---

> > ### Author Response · Authors · 2025-11-21
> >
> > > 3. There could be obvious discrepancy between the synthetic training data and the real-world tool use cases. Connected subgraphs are collected to form the synthetic training data as introduced in Section 3.1, but in practice there could be tool calling trajectory on a subgraph with much more nodes (5+ steps v.s. 4 API nodes in this work) and even more some query could involve multiple subgraphs.
> >
> > Thank you for raising this point.
> >
> > **1. Why we limit synthetic subgraphs to 4 nodes .**
> >
> > We set the subgraph size to a reasonable range based on our analysis of API dependency graphs; and excessively long API chains are unrealistic for a single user query. Consistent with this intuition, both RestBench and API-Bank contain gold solution paths with at most four API functions. Thus, a 4-node cap provides full coverage of real trajectories in our evaluation settings. Importantly, this is not a limitation of GOAT—the framework can generate longer chains simply by increasing this hyperparameter when applied to APIs with deeper dependency structures.
> >
> > **2. Handling real queries that involve multiple subgraphs.**
> >
> > We agree that some real-world user queries may involve multiple independent tool-use chains—e.g., two coordinated clauses that describe separate subtasks. Although our benchmarks do not include queries requiring multiple disconnected subgraphs, GOAT can naturally extend to such cases. A complex user query can be decomposed into subtasks as in [3],  each grounded on its own connected subgraph. Because GOAT already trains the model to execute each subgraph correctly, adding a subtask-decomposition prompt at inference time would allow the model to handle multi-subgraph queries without modifying the training pipeline. We consider exploring such multi-subgraph settings a promising direction for future work.
> >
> > [3] Khot, Tushar, et al. "Decomposed Prompting: A Modular Approach for Solving Complex Tasks." The Eleventh International Conference on Learning Representations.
> >
> > &nbsp;&nbsp;
> > &nbsp;&nbsp;&nbsp;
> >
> > > 4. Lack of comparison with recent closed-source models like GPT-4o or Claude 4 sonnet. Given that GPT-4o has strong zero-shot tool-use capabilities, omitting this comparison weakens the claim that GOAT-trained open-source models are "competitive with closed-source systems."
> >
> > Thank you for pointing this out. To address this concern, we additionally evaluated **GPT-4o**, a strong recent closed-source model, across all three benchmarks (API-Bank, RestBench, and GOATBench) and compared its zero-shot performance with GOAT-trained open-source models. As shown in the tables below, GOAT-fine-tuned Llama models show comparable performance with GPT-4o, even outperforming on some metrics—demonstrating that GOAT-trained open models remain competitive with closed-source systems.
> >
> > **RestBench**
> >
> > | Backbone       | Training Method | TMDB Success % | Spotify Success % |
> > |-------------|------------------|----------------|--------------------|
> > | Llama2-13B  | Zero-shot                | 0              | 3.5                |
> > | Llama2-13B  | GOAT             | 7              | **28.1**           |
> > | Llama3-8B  | Zero-shot                | 1              | 10.5               |
> > | Llama3-8B   | GOAT             | **14**         | 19.3               |
> > | GPT-4o      | Zero-shot                | 11             | 17.5               |
> >
> > **API-Bank (Plan+Retrieve+Call)**
> >
> > | Backbone      | Training Method | Success % |
> > |------------|------------------|-----------|
> > | Llama-7B   | Zero-shot             | 0        |
> > | Llama-7B   | GOAT             | 38        |
> > | Llama3-8B  | Zero-shot             | 34    |
> > | Llama3-8B  | GOAT             | **64**    |
> > | GPT-4o     | Zero-shot                | 60        |
> >
> > **GOATBench**
> >
> > | Backbone      | Training Method | Inter Tool SA | Inter Tool IA | Inter Tool SR | Single Tool SA | Single Tool IA | Single Tool SR |
> > |------------|------------------|----------------|----------------|----------------|-----------------|-----------------|-----------------|
> > | Llama3-8B  | Zero-shot                | 10.4           | 3.3            | 4.1            | 18.6            | 6.0             | 7.1             |
> > | Llama3-8B  | GOAT             | **59.0**       | **26.1**       | 14.4           | **68.9**        | **35.6**        | 24.5            |
> > | GPT-4o     | Zero-shot                | 17.7           | 5.5            | **22.2**       | 35.5            | 10.6            | **27.8**        |
> >
> > *All experiments use the Baseline prompting method.

---

> > > ### Author Response · Authors · 2025-11-21
> > >
> > > > 5. Missing technical details such as the cost of API calls, number of training data points for LLM fine-tuning and number of query-document pairs for SBERT training.
> > >
> > > Thank you for the helpful comment. We added the requested details to Appendix B of the experiment setup.
> > >
> > > **1. Cost of API calls.**
> > >
> > > We clarify that API calls arise in two places:
> > > (1) **during dependency-graph construction** (≈2 API calls per edge in step 3), and
> > > (2) **during synthetic data generation** (≈one API call per node in each sampled subgraph).
> > > All APIs used in our experiments (TMDB, Spotify, StableToolBench [4]) are free and do not incur monetary charges, so the cost is reflected only in computation time rather than price. If the reviewer was referring to monetary cost, we confirm that no paid API endpoints were used.
> > >
> > > **2. Number of LLM training instances.**
> > >
> > > The number of generated training data points for LLM fine-tuning is as follows:
> > > | Dataset                     | # Training Instances |
> > > |-----------------------------|----------------------|
> > > | RestGPT–TMDB                | 8,570                |
> > > | RestGPT–Spotify             | 924                  |
> > > | API-Bank (Plan+Retrieve+Call)                    | 108                  |
> > > | GOATBench–Entertainment     | 1,631                |
> > > | GOATBench–Financial         | 1,354                |
> > > | GOATBench–Food              | 650                  |
> > > | GOATBench–Travel            | 420                  |
> > >
> > >
> > > **3. Number of query–document pairs used for SBERT training.**
> > >
> > > | Dataset                     | # Query–Document Pairs |
> > > |-----------------------------|-------------------------|
> > > | RestGPT–TMDB                | 33,169                  |
> > > | RestGPT–Spotify             | 3,389                   |
> > > | API-Bank (Plan+Retrieve+Call)                   | 180                     |
> > > | GOATBench–Entertainment     | 5,091                   |
> > > | GOATBench–Financial         | 4,752                   |
> > > | GOATBench–Food              | 1,957                   |
> > > | GOATBench–Travel            | 1,166                   |
> > >
> > >
> > > [4] Guo, Zhicheng, et al. "Stabletoolbench: Towards stable large-scale benchmarking on tool learning of large language models." arXiv preprint arXiv:2403.07714 (2024).
> > >
> > > &nbsp;&nbsp;
> > > &nbsp;&nbsp;&nbsp;
> > >
> > > > 6. Section 3.1.3 seems like discussion on related works, then it would be better to move this paragraph to section 2.
> > >
> > > We appreciate the reviewer’s suggestion. However, we would like to clarify why we chose to present this discussion separately. Section 3.1.3 was not meant to serve as a related-work comparison but rather as a methodological clarification of  why a call-first formulation is needed, and how GOAT enables efficient benchmark construction with minimal human effort—points that would be difficult to convey through a short mention in the related-work section. A simple or brief comparison in the related work section can easily lead to confusion, so we believed it was important to explicitly highlight the differences.
> > >
> > > That said, we agree that some of the higher-level comparisons can be placed earlier for better organization. We will move the broad contextual comparison to Section 2 and refine Section 3.1.3 to focus specifically on our core contributions while keeping the section concise and contribution-focused.

---

> ### Comment · Reviewer_4MzE · 2025-11-25
>
> Thanks for the authors' responses. Most of my doubts were addressed, but I have a further concern in the added experiment.
> Why GOAT trained open-source models perform inconsistently in three metrics SA/IA/SR on GOATBench comparing with GPT-4o? The experimental results show that the GPT-4o gains very low Inter Tool IA but much higher Inter Tool SR, which seems to tell that GPT-4o can execute task successfully without ensuring correct match of arguments/function name in API calling. Since my concern about the main claim has not been resolved yet, I determine to keep my original rating.

---

### Official Review · Reviewer_e8gR · 2025-10-31

**Soundness:** 3
**Presentation:** 3
**Contribution:** 3
**Rating:** 6
**Confidence:** 3

**Summary:**

This paper proposes GOAT, a human annotation-free training framework for large language model (LLM) agents operating in goal-oriented tool-use scenarios, specifically for planning and executing interdependent API calls given only API documentation.  The approach consists of automatically generating synthetic datasets by parsing API docs, constructing and refining API dependency graphs, sampling realistic interdependent API task sequences, and fine-tuning LLM agents and retrievers on these synthetic tasks.  The paper introduces a new benchmark, GOATBench, and empirically demonstrates that models trained with GOAT achieve strong, often state-of-the-art, performance on multiple goal-oriented benchmarks, sometimes surpassing closed-source models.

**Strengths:**

1. This paper proposes a comprehensive, multi-stage, and meticulous automatic data generation method for constructing goal-oriented API execution datasets from unlabeled API documentation. The entire process is carefully decomposed into multiple stages—embedding-based similarity filtering, LLM semantic verification, and real API call execution—significantly improving the reliability of the dependency graph while maintaining computational feasibility.
2. A novel benchmark is proposed: GOATBench is a human-validated, non-trivial benchmark set for goal-oriented tool use, covering multiple domains, and specifically designed to test the generalization ability of models on combinations of seen and unseen APIs.

**Weaknesses:**

1. Additional scaling experiments could be included, and all current experiments appear to have been run only once without reporting standard deviations or confidence intervals, which limits statistical reliability.
2. Although GOATBench is described as a comprehensive goal-oriented benchmark, its construction pipeline is largely reused from the training process, raising potential risks of data leakage or overfitting to generation biases, particularly for the “seen” API compositions.
3. The introduction of GOATBench is a valuable contribution, and Appendix D provides useful implementation details. However, the paper does not sufficiently describe the data characteristics and distribution, nor does it relate GOATBench to existing benchmark efforts in this area. The authors are encouraged to enrich Appendix D with a more detailed comparison and statistical summary, referencing representative benchmark papers for context.

**Questions:**

see weaknesses.

---

> ### Author Response · Authors · 2025-11-21
>
> We thank the reviewer for the thoughtful feedback and suggestions. We address each concern point-by-point below.
> ​
> &nbsp;&nbsp;
> &nbsp;&nbsp;&nbsp;
>
> > 1. Additional scaling experiments could be included, and all current experiments appear to have been run only once without reporting standard deviations or confidence intervals, which limits statistical reliability.
>
> We appreciate the reviewer’s concern regarding statistical reliability. We report multi-run results on GOATBench to show statistical significance. The non-finetuned baseline is deterministic, while GOAT-trained models are fine-tuned with **five**-different random seeds.  For each metric, we perform a **two-sided paired t-test** comparing fine-tuned models against the non-finetuned baseline, and we report the resulting p-values. As shown in the table below, GOAT-trained models outperform the baseline across all metrics, with **most achieving p < 0.01** and the remaining two achieving p < 0.05.
>
> We will additionally run the same statistical test for all remaining tables and include the full p-value results in the Appendix.
>
> &nbsp;&nbsp;
> &nbsp;&nbsp;&nbsp;
>
> || Inter Tool SA | Inter Tool IA | Inter Tool SR | Single Tool SA | Single Tool IA | Single Tool SR |
> |-------------------|---------------|--------|--------|----------------|--------|--------|
> | **Zero-shot** | 10.4          | 3.3    | 4.1    | 18.6           | 6.0    | 7.1    |
> | **GOAT finetuned Mean**| 57.3          | 20.5   | 11.7   | 65.5           | 23.6   | 18.6   |
> | **p-value**       | 2.6e-07       | 2.2e-03| 5.6e-03| 2.2e-05        | 3.3e-02| 1.6e-02|
>
> *All experiments in this table use the Baseline prompting method with a Llama-3-8B backbone.
>
> &nbsp;&nbsp;
> &nbsp;&nbsp;&nbsp;
>
> > 2. Although GOATBench is described as a comprehensive goal-oriented benchmark, its construction pipeline is largely reused from the training process, raising potential risks of data leakage or overfitting to generation biases, particularly for the “seen” API compositions.
>
>
> We appreciate the reviewer’s concern. Although GOATBench is built using the same API dependency graph as the training pipeline, its design explicitly mitigates leakage from training instances and still evaluates genuine generalization ability.
>
> **1. Consistent trends with fully human generated benchmarks.**
>
> If GOATBench were benefiting from construction biases, we would expect to see gains that are specific to GOATBench but not to external human-generated benchmarks. Instead, we observe the same qualitative improvements on external goal-oriented benchmarks—RestBench [1] and API-Bank [2]—whose test sets are fully human-authored. This consistency indicates that GOAT improves general goal-oriented reasoning rather than exploiting artifacts specific to GOATBench.
>
> **2. GOATBench’s seen/unseen split is designed to test generalization.**
>
> **Unseen split**: every test case includes at least one API never observed during training, forcing the model to reason over new APIs and compositions rather than memorized patterns.
>
> **Seen split**: while built from the same API inventory, we explicitly avoid instance-level leakage by (i) constraining initial argument values to never overlap with training and (ii) applying human editing to paraphrase user queries and remove unnatural or spurious API paths.
> Our use of a shared synthetic pipeline for both training and evaluation follows prior tool-learning frameworks such as ToolLLM [3] and Gorilla [4], which similarly build benchmarks from the same generation procedures used to create their training data.
>
> &nbsp;&nbsp;
> &nbsp;&nbsp;&nbsp;
>
> [1] Song, Yifan, et al. "Restgpt: Connecting large language models with real-world restful apis." arXiv preprint arXiv:2306.06624 (2023).
>
> [2] Li, Minghao, et al. "API-Bank: A Comprehensive Benchmark for Tool-Augmented LLMs." Proceedings of the 2023 Conference on Empirical Methods in Natural Language Processing. 2023.
>
> [3] Qin, Yujia, et al. "ToolLLM: Facilitating Large Language Models to Master 16000+ Real-world APIs." ICLR. 2024.
>
> [4] Patil, Shishir G., et al. "Gorilla: Large language model connected with massive apis." Advances in Neural Information Processing Systems 37 (2024): 126544-126565.

---

> > ### Author Response · Authors · 2025-11-21
> >
> > > 3. The introduction of GOATBench is a valuable contribution, and Appendix D provides useful implementation details. However, the paper does not sufficiently describe the data characteristics and distribution, nor does it relate GOATBench to existing benchmark efforts in this area. The authors are encouraged to enrich Appendix D with a more detailed comparison and statistical summary, referencing representative benchmark papers for context.
> >
> > Thank you for the helpful suggestion. We added a detailed statistical comparison of GOATBench, RestBench, and API-Bank to Appendix D, including domain coverage, number of APIs, number of instances, and average path length, as well as the distribution of dependency-chain lengths. We believe this summary clarifies the characteristics of GOATBench and its relation to prior benchmarks.
> >
> > &nbsp;&nbsp;
> > &nbsp;&nbsp;&nbsp;
> >
> > |                  | GOATBench | RestBench | API-Bank |
> > |------------------|-----------|-----------|----------|
> > | # of Domains     | 4         | 2         | 8        |
> > | # of APIs        | 182       | 94        | 73       |
> > | # of Instance    | 747       | 157       | 314      |
> > | Avg. Len         | 3.1       | 2.41      | 2.91     |
> >
> > &nbsp;&nbsp;
> > &nbsp;&nbsp;&nbsp;
> >
> > | Subgraph size | # of Instance |
> > |---------------|---------------|
> > | 2             | 90            |
> > | 3             | 480           |
> > | 4             | 163           |

---

> > ### Comment · Reviewer_e8gR · 2025-11-24
> >
> > Thanks for your response. As the score is already very high, I have decided to maintain it as is.

---

### Note · Authors · 2025-12-02

I have read and agree with the venue's withdrawal policy on behalf of myself and my co-authors.